# Dynamic interaction of BRCA2 with telomeric G-quadruplexes underlies telomere replication homeostasis

Junyeop Lee [1,3], Keewon Sung [2,3], So Young Joo [1], Jun-Hyeon Jeong[1], Seong Keun Kim [2✉] & Hyunsook Lee [1✉]

BRCA2-deficient cells precipitate telomere shortening upon collapse of stalled replication forks. Here, we report that the dynamic interaction between BRCA2 and telomeric G-quadruplex (G4), the non-canonical four-stranded secondary structure, underlies telomere replication homeostasis. We find that the OB-folds of BRCA2 binds to telomeric G4, which can be an obstacle during replication. We further demonstrate that BRCA2 associates with G-triplex (G3)-derived intermediates, which are likely to form during direct interconversion between parallel and non-parallel G4. Intriguingly, BRCA2 binding to G3 intermediates promoted RAD51 recruitment to the telomere G4. Furthermore, MRE11 resected G4-telomere, which was inhibited by BRCA2. Pathogenic mutations at the OB-folds abrogated the binding with telomere G4, indicating that the way BRCA2 associates with telomere is innate to its tumor suppressor activity. Collectively, we propose that BRCA2 binding to telomeric G4 remodels it and allows RAD51-mediated restart of the G4-driven replication fork stalling, simultaneously preventing MRE11-mediated breakdown of telomere.

[1] Department of Biological Sciences & IMBG, Seoul National University, Seoul 08826, South Korea. [2] Department of Chemistry, Seoul National University, Seoul 08826, South Korea. [3]These authors contributed equally: Junyeop Lee, Keewon Sung. ✉email: seongkim@snu.ac.kr; HL212@snu.ac.kr

Linear eukaryotic chromosomes experience 'end replication problem'. This means that the protective effect of telomeres is lost over time, resulting in decreased genome integrity. Differentiated cells usually lack telomerase activity, and thus face senescence due to progressive telomere shortening. Overcoming cellular senescence is a critical step during cell transformation in tumorigenesis. Therefore, telomerase activity is observed in majority of cancer cells. Alternatively, approximately ~10–15% of cancers exhibit ALT (Alternative Lengthening of Telomeres), an alternate mechanism that maintains and elongates telomeres independent of telomerase activity, allowing cancer cells to thrive and proliferate.

Telomere repeat array sequences are well conserved from yeast [$TG_{2-3}(TG)_{1-6}$] to human (TTAGGG). Notably, consecutive guanine repeats can interact through non-canonical Hoogsteen base-pair formation, where the guanine becomes both the donor and the acceptor of hydrogen, instead of engaging in Watson-Crick pairing. As a result, Hoogsteen pairs between four guanines can form a flat plane called a G-tetrad[1]. Two or more G-tetrads can stack to form a non-canonical secondary structure called a G-quadruplex (G4)[2]. Intriguingly, single-stranded G-rich telomeric DNA is also able to form G4 structures in a physiological environment[3]. More recently, G4 formation at telomeres has been directly visualized in human cells[4]. During replication of the lagging strand at telomeres, the single-stranded telomeric repeat can potentially fold into G4. If left unresolved, G4s can cause telomere erosion and result in aberrant recombination[5]. At the same time, single-stranded telomere folding into G4s can control their accessibility to telomerase, thereby regulating telomere maintenance[6,7].

Germ-line mutation of the breast cancer susceptibility gene, *BRCA2*, predisposes carriers to cancers of the breast, pancreas, ovary, and other tissues. We and others have shown that disruption of *BRCA2* results in telomere shortening[8,9]. MRE11 functions by resecting DNA of stalled replication forks, and it is possible that BRCA2 protects telomeres from this potentially damaging resection activity. Here, we examined the mechanism by which BRCA2 maintains telomere replication homeostasis. Using molecular and biophysical methods, we show that the unique mode of BRCA2 interaction with telomere G4 protects G4-driven stalled replication forks from nuclease attack, simultaneously facilitating the restart of stalled forks.

## Results

### BRCA2 binds to telomeric G-quadruplex structures.
We previously observed that BRCA2 localizes to ~10 % of telomeres during S phase[9]. Here, we used immunoFISH to confirm that BRCA2 localizes to telomeres in HeLa cells (NFLAP-BRCA2-HeLa[10,11]) as well (Supplementary Fig. 1). Absence of BRCA2 results in fragile telomeres, which led us to speculate that BRCA2 may be critical in telomere maintenance[9]. To unveil the mechanistic role that BRCA2 plays in telomere replication, we investigated the molecular nature of BRCA2 binding to S phase telomeres.

BRCA2 is a ~384 kDa protein with no apparent homology to other known proteins. However, there are three oligonucleotide/oligosaccharide-binding folds (OB-folds) at the C-terminus, which show limited homology to the OB-fold of RPA. Notably, POT1, a member of the telomeric Shelterin protein, specifically binds to single-stranded telomeric DNA and also possesses OB-folds[12–14]. The BRCA2 DNA-binding domain (DBD), which contains the OB-folds, can bind to single-stranded DNA in complex with the cofactor DSS1 (Deleted in split-hand/split-foot syndrome protein). This complex coordinates homologous recombination (HR) during double-stranded DNA break (DSB) repair. The structure of the DBD in complex with DSS1 and a random ssDNA has been solved[15].

Considering these informations into account, we asked whether BRCA2 exerts telomere binding activity without DSS1. In order to test this idea, three GST-tagged proteins, encompassing the DBD at the C-terminus of human BRCA2 (B2-7, B2-8, B2-9)[11] were purified. Binding of these recombinant proteins to radio-labeled G-rich single-strand or C-rich single-strand oligonucleotides representing telomeric sequences was tested using EMSA (Electrophoretic Mobility Shift Assay). While none of the recombinant fragments associated with the C-rich strand, the OB2 and OB3-containing species (B2-8) specifically bound to the G-rich telomere sequence (Supplementary Fig. 2).

To further extend the study, MBP (maltose-binding protein)-tagged BRCA2OB (2665-3197 a.a.), which contains OB-folds but lacks helical domain, was purified from *E. coli* (Fig. 1a) and subjected to EMSA. Single-stranded G-rich telomere arrays [TelG5; (GGTTAG)$_5$], single-stranded C-rich arrays [TelC5; (CTAACC)$_5$], or double stranded oligonucleotides (TelG5:TelC5) were end-labeled with γ-$^{32}$P-[ATP] then employed in EMSA (Fig. 1b and Supplementary Table 1). The OB-folds domain of BRCA2 (BRCA2OB) specifically bound to single-stranded G-rich telomere repeat array (TelG5), but not to single-stranded C-rich (TelC5) nor to double-stranded telomere repeats (Fig. 1b; Supplementary Table 1). The binding of BRCA2OB to TelG5 increased in a dose-dependent manner (Fig. 1c, lanes 1-4). Furthermore, adding unlabeled cold TelG5 abolished formation of the BRCA2OB-TelG5 complex (Fig. 1c, lane 5) while non-related poly (dI-dC) did not (lane 6), confirming that the binding of BRCA2OB to TelG5 is specific.

In TelG5, four consecutive TTAGGG repeat arrays exist with GG and TTAG flanking sequences at the 5'- and 3'-end, respectively (Supplementary Table 1). Therefore, TelG5 can form G4. We asked whether BRCA2OB can interact with this telomeric G4. For this, increasing numbers of telomere repeat arrays were tested for binding to BRCA2OB. BRCA2OB bound to single-stranded G-rich oligonucleotides with more than 4 repeats but not to those with less (Fig. 1d): slight binding of BRCA2OB was observed in TelG4, which can form tetrads but not G4; binding was most apparent in TelG5 and TelG6, which can fold into G4. BRCA2OB did not associate with TelG3, which may fold into secondary structure but cannot stack to form G4 (Fig. 1d).

Notably, BRCA2OB associated with TelG5 in the absence of DSS1. This suggests that the nature of BRCA2 binding to G-rich telomeres is distinct from non-sequence-specific BRCA2-DSS1 interaction with single-stranded DNA during HR[15,16]. To test this idea, we utilized a mutant construct termed TelG5GAG (TTAGAG repeat), where GGG in TelG5 is replaced with GAG to disrupt folding into G4 structures. Circular dichroism (CD) spectra confirmed that TelG5GAG is mainly composed of unfolded single-stranded DNA (Supplementary Fig. 3). By comparing the interaction of BRCA2OB with TelG5 and TelG5GAG, we confirmed that BRCA2OB associates with G4 telomeres, but not single-stranded DNA (Fig. 1e).

Next, we examined the effect of the G4-stabilizing ligand pyridostatin (PDS)[17,18] on the BRCA2OB-TelG5 interaction. PDS is a potential anti-cancer drug because it induces DNA damage[19–21]. PDS-bound G4 telomeres are likely to remain unresolved, and then undergo resection by nucleases, which would be lethal to cancer cells. When PDS was pre-incubated with TelG5, it inhibited the BRCA2OB-TelG5 interaction in a dose-dependent manner (Fig. 1f). Together, these results suggest that BRCA2OB specifically interacts with G4 at telomeres.

### Unique mode of BRCA2 interaction with telomeric G4.
Other than telomere, sequences that can form G4 are found throughout

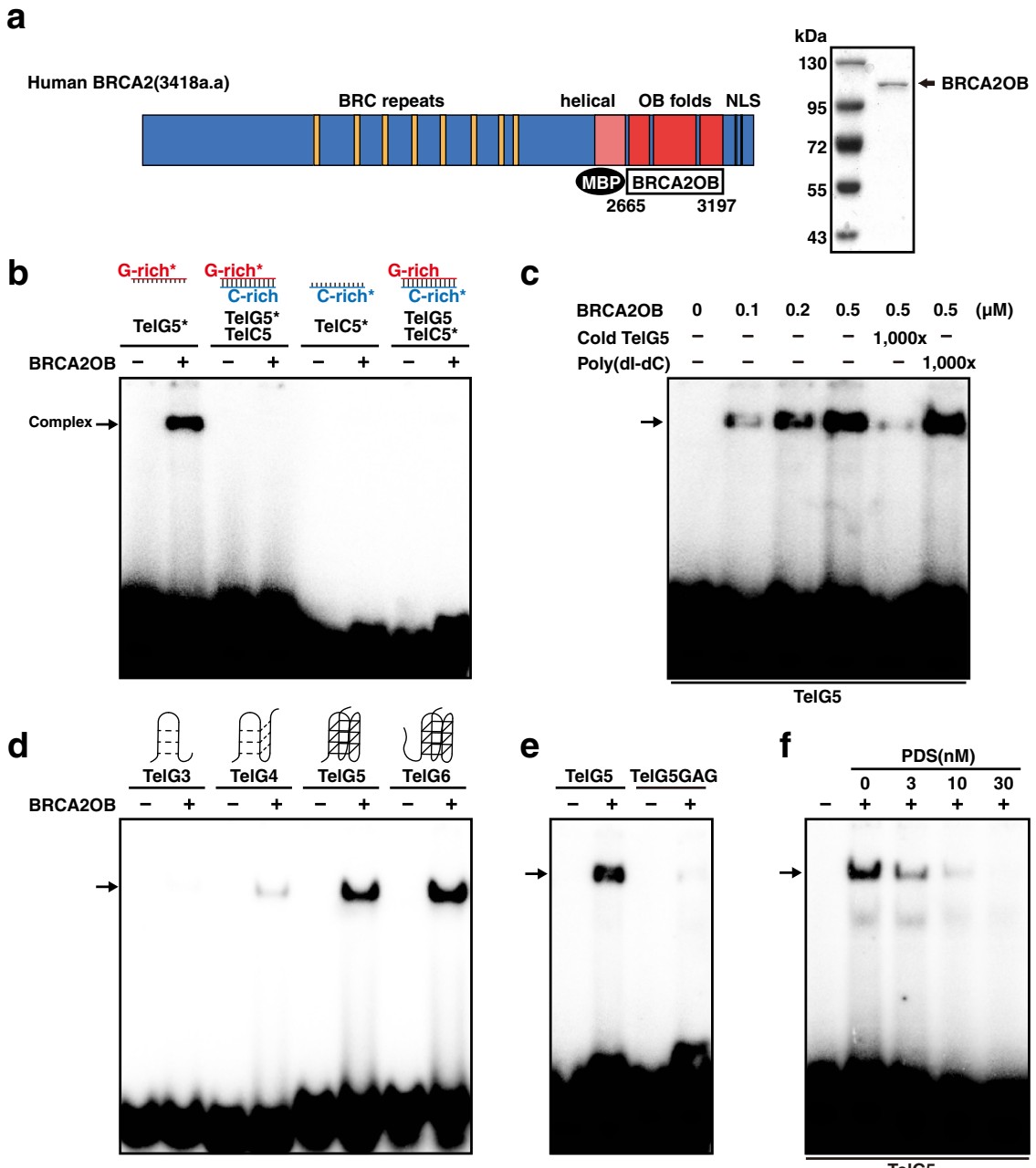

**Fig. 1 The BRCA2 OB-folds binds to telomeric G-quadruplexes. a** Schematic illustration of human BRCA2. BRC repeats, OB folds, and the nuclear localization signal (NLS) are marked. Recombinant BRCA2 OB-folds domain (2665-3197 a.a.; BRCA2OB), tagged with N-terminal MBP was purified (SDS-PAGE gel at right). **b** Electrophoresis mobility shift assay (EMSA) was used to assess binding of recombinant BRCA2OB to telomere repeats. TelG5, G-rich single-stranded (GGTTAG)$_5$ repeats; TelC5, C-rich strand complementary to TelG5; TelG5:TelC5, double-stranded. $^{32}$P-labeled hot probe is marked with an asterisk(*). Arrow marks the BRCA2OB-bound hot probe. **c** Increasing concentrations of BRCA2OB were incubated with TelG5 (lanes 2 to 4). Binding specificity was confirmed by competing with 1000X excess of unlabeled TelG5 (+, Cold TelG5, lane 5) and Poly(dI-dC) for negative control (lane 6). **d** Single-stranded (ss) telomeric repeat oligonucleotides of increasing length (TelG3 – TelG6, see illustration at the top) were incubated with (+) or without (-) BRCA2OB and subjected to EMSA. The binding of BRCA2OB to telomere repeats was detected from TelG5 and TelG6. **e** BRCA2OB specifically binds to TelG5, which forms the G-quadruplex (G4), but not to the single-stranded mutant (TelG5GAG). **f** A G4 ligand PDS challenges the binding of BRCA2OB to G4 in a dose-dependent manner. EMSA analyses were performed in the presence of 100 mM KCl. All experiments were repeated more than three times independently.

the genome, especially at promoters[22,23] including the *c-MYC* promoter[24]. Unlike BRCA1, the likelihood of BRCA2 being involved in transcriptional control is low. We asked if BRCA2OB binding discriminates telomere G4 from non-telomeric G4. To test this, we compared BRCA2OB binding to telomeric G4 sequence versus the *c-MYC* promoter or an artificially designed oligonucleotide that has a G4-forming sequence motif (Supplementary Table 1). In EMSA, we found that BRCA2OB specifically associated with telomere G4; it bound markedly less to *c-MYC* promoter DNA or the artificially designed G4, while interaction with telomere G4 was significant (Supplementary Fig. 4).

Although mammalian telomeres are mostly composed of TTAGGG repeat arrays, variations in the sequence exist at the proximal or subtelomeric regions as well as within some telomere regions. These variants are referred to as telomere variant repeats (TVRs): TCAGGG, TGAGGG, CTAGGG, TTGGGG, and TTTGGG[25–29]. Notably, variations in TVRs are restricted to the non-guanine loop, leaving intact with the consecutive guanine repeats that are responsible for stacking square-planar tetrads. This implies that the ability to fold into G4 may be a critical feature that is a defining characteristic of telomeres.

In EMSA, BRCA2OB binding was observed to TelG (typical TTAGGG), TelG-TCA (TCAGGG), and TelG-TGA (TGAGGG) variants. In comparison, binding of BRCA2OB to TelG-CTA (CTAGGG) and the unfolded mutant control TelG-GAG was not detected (Supplementary Fig. 5 and Supplementary Table 1).

In order to assess the binding affinity of BRCA2OB for G4-forming telomeres, we performed EMSA and roughly estimated the dissociation constant ($K_d$) of BRCA2OB for three different telomeric G4 sequences (Fig. 2a). BRCA2OB showed a $K_d$ of ~4.4 µM of for TelG5, $K_d = $ ~456 nM to TelG5TTT, and $K_d = $ ~1.4 µM to TelG5T2G4 (Fig. 2a). The $K_d$ of POT1 binding to G-rich ssDNA representing telomeres is 38.2 nM[30]; TLS/FUS to telomeric G4 is 37 nM[31]; and nucleolin to telomeric G4 is 2.574 µM[32].

Notably, BRCA2OB binding affinity increased ~9 times and ~3 times more respectively in assays using TelG5TTT (TTTGGG repeats) and TelG5T2G4 (TTGGGG repeats), compared to TelG5 (Fig. 2b). These sequences represent the telomeres of ciliates[33] and Tetrahymena[34,35], respectively, and are also found in cancers[36]. Next, we compared the binding of the OB folds from the Shelterin complex protein POT1 (POT1OB) to BRCA2OB. Notably, G4-forming property of telomere overhang enhances POT1/TPP1 binding, preventing from DNA damage-inducing RPA to load on ssDNA telomere[37]. Furthermore, POT1/TPP1 first captures G4, but the increased binding of POT1/TPP1 destabilizes telomere G4[38]. The binding mode of POT1 to telomere variants contrasted with that of BRCA2OB (Fig. 2b). POT1OB bound to TelG5 (TTAGGG repeats) strongly as expected, but it did not bind to the other variants that BRCA2OB interacts well with (Fig. 2b), indicating that the mode of BRCA2 binding to telomere differs from that of POT1.

**Direct observation of G4 dynamics and BRCA2 interaction.** Telomere G4 can form various topologies including parallel, anti-parallel, and hybrid conformations[39–42], depending on the strand polarities, the number of guanine repeats, the length and sequence of the non-guanine loop, the concentration and type of monovalent metal cations, and molecular crowdedness[43–45]. In addition, single-molecule analysis has identified intrinsic structural rearrangements between three detectable conformations (i.e., the parallel (P), non-parallel (NP), and unfolded (UF)), indicating the dynamic nature of G4[43,46,47].

To reveal the molecular characteristics of the interaction between BRCA2 and telomeric G4 in detail, we utilized single-molecule FRET (smFRET) spectroscopy. A single-stranded oligonucleotide of TelG5 was partially duplexed with stem having a short sequence. The 3'-end of TelG5 was labeled with the FRET donor (Cy3) and the opposite 5'-end of the stem was labeled with the acceptor (Cy5) (Fig. 2c and Supplementary Table 2). This oligonucleotide was immobilized on a quartz surface for direct observation of conformational dynamic shifts between UF, NP, and P conformations (Fig. 2c).

For reference, we measured the circular dichroism (CD) spectra of TelG5 under K$^+$-rich conditions, which showed a

major peak at ~290 nm and a positive signal at ~265 nm (Fig. 2d). According to previous reports, positive peaks at 290 nm and 265 nm in CD spectra of G4 correspond to the characteristics of NP and P folding, respectively[42]. In the smFRET histogram, TelG5 exhibited two peaks in the presence of 100 mM KCl: a major middle-FRET state ($E = 0.47 \pm 0.006$; s.d., $n = 5$) and a minor high-FRET state ($E = 0.66 \pm 0.015$; s.d., $n = 5$) (Fig. 2e). Correlating with the CD spectra, we assigned the major middle-FRET state as NP and the minor high-FRET state as P conformation (Fig. 2e, f). A small fraction of the low-FRET state ($E = 0.24 \pm 0.014$; s.d., $n = 5$) was assigned as unfolded (UF), because it was identical to the smFRET value of TelG5GAG (Supplementary Fig. 6). Although there have been conflicting reports on the structural assignment of the middle- and high-FRET states, some are consistent with ours[37,47–49], while others propose the middle-FRET as P conformation and high-FRET as NP conformation[43,46]. We note that the discrepancy is observed in G4 constructs without single-stranded tail flanking the G4. However, for G4 constructs possessing flanking sequences as in TelG5, our FRET-state assignments are consistent with previous studies[37,49]. In single-molecule time trajectories, 48% ± 0.7% (s.d., $n = 3$) of TelG5 molecules showed frequent transitions among the UF, NP, and P conformations in 100 mM KCl buffer, confirming that telomeric G4 is highly dynamic and heterogeneous in nature (Fig. 2f).

To directly monitor the BRCA2-telomere G4 interaction, we added 1 µM of recombinant BRCA2OB to the surface-immobilized TelG5 DNA. As a result, a new FRET peak appeared ($E = 0.57 \pm 0.006$; s.d., $n = 4$) between the NP and P conformation peaks (Fig. 2g). The rise of the new peak with concurrent decrease in the NP and P population became apparent in the differential density histogram (Fig. 2h), where the FRET histogram upon BRCA2OB incubation (Fig. 2g) was subtracted by that of TelG5 alone (Fig. 2e). The new FRET state was consistently identified in single-molecule time trajectories upon addition of BRCA2OB, and was observed right after the FRET transitions between the NP and P conformations (Fig. 2i). These data indicate that the new FRET state arises from a BRCA2OB-bound (BD) conformation (Fig. 2g, i, BD state). Importantly, these results corroborate the interaction between BRCA2 and the dynamic telomere G4 at the single-molecule level. The data also suggest that BRCA2OB may recognize one of the telomeric G4 structures: P, NP conformations and other potentially short-lived states.

**Biophysical properties of telomeric G4 dictate BRCA2 binding.** To further analyze the characteristics of the interaction between BRCA2OB and the G4 telomere, we compared the biophysical properties of TelG5 with the three previously characterized G4 variants (TTA to TAA for TelG5TAA; TTT for TelG5TTT; TT for TelG5TT (Supplementary Table 2)) using smFRET spectroscopy[43]. The relative fraction of the P conformation in smFRET markedly increased in TelG5TTT (46% ± 3%, $n = 5$) and TelG5TT (53% ± 2%, $n = 2$), compared to TelG5 (37% ± 3%, $n = 5$) and TelG5TAA (23% ± 1%, $n = 3$), as reported[43]. The UF (unfolded) fraction was highest in smFRET data from TelG5TAA (11% ± 0.4%, $n = 3$) (Fig. 3a). Similar results were obtained from the CD spectra (Supplementary Fig. 7a).

Upon addition of BRCA2OB to these variants, all four substrates exhibited increased differential density in the region between the NP and P conformational FRET value (i.e., BD state) as well as the reduction of the NP and P population (Fig. 3b and Supplementary Fig. 7b,c). Moreover, the sum of the positive differential density, which partially reflects the fraction of the BRCA2OB-bound (BD) state was lower in TelG5TAA

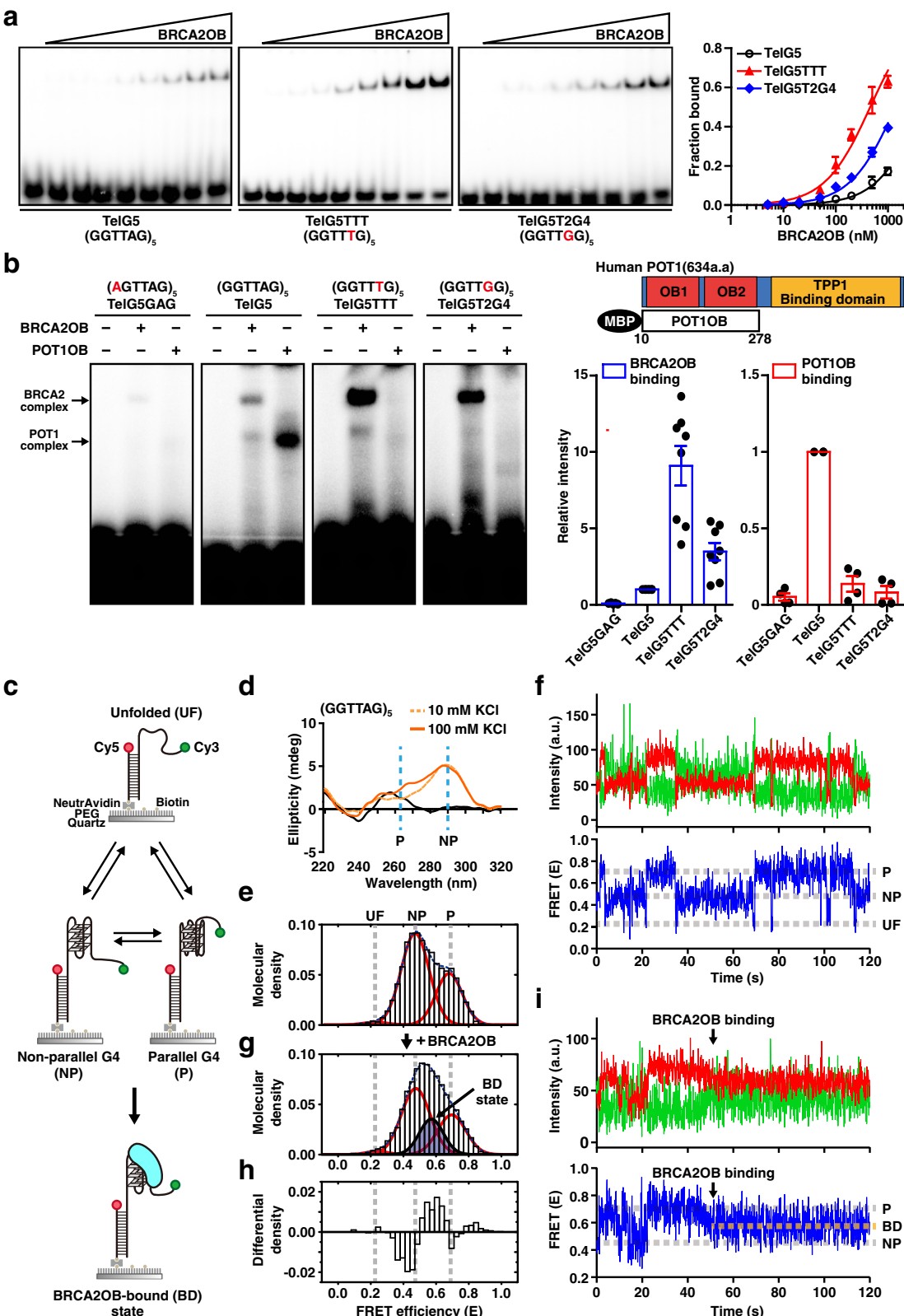

and TelG5. In comparison, BD fraction increased in TelG5TTT and TelG5TT when 100 nM (Fig. 3b) and 1 µM of BRCA2OB (Supplementary Fig. 7c) were added. To quantify the BRCA2OB binding propensity of the four variants, EMSA was performed. It was apparent that TelG5TTT and TelG5TT retained the strongest binding to BRCA2OB: 9- and 13-fold more, respectively, compared to TelG5 (Fig. 3c). These results led us

to investigate the nature of various telomeric G4 structures in association with different BRCA2OB-binding properties.

First, the individual time trajectories of FRET transitions were analyzed by calculating the dwell time in each conformation (Supplementary Fig. 8). The average dwell time in the UF conformation was the highest and four-times longer in TelG5TAA (4.3 s). TelG5TAA showed the weakest interaction,

**Fig. 2 Direct observation of BRCA2 binding to telomere G4 at the single-molecule level. a** TelG5, TelG5TTT, and TelG5T2G4 probes (10 nM) were incubated with BRCA2OB at the following concentrations: 0, 5, 10, 20, 50, 100, 200, 500, and 1000 nM (left). The $K_d$ values were analyzed by curve-fitting using nonlinear regression (mean ± s.d.) (Right). TelG5, $K_d = 4.4$ µM; TelG5TTT, $K_d = 456$ nM; TelG5T2G4, $K_d = 1.4$ µM. The result is from more than three independent experiments. **b** Purified BRCA2OB and POT1OB were subjected to EMSA in the presence of 100 mM KCl with G4-forming telomere variant repeats (TVRs); TelG5, TelG5TTT, TelG5T2G4, and unfolded mutant TelG5GAG. Human POT1 and the POT1 OB-fold domains (POT1OB) are illustrated at top right. The OB-folds and TPP1 binding domain are marked. Bar graph, binding intensity relative to TelG5 in EMSA (mean ± s.e.m.). The experiments were repeated at least four times independently. **c** Schematic drawing of the single-molecule FRET (smFRET) assay with dual-labeled TelG5 and BRCA2OB. The G4 conformational dynamics between the unfolded (UF), non-parallel (NP), and parallel (P) folded conformations, and BRCA2OB-bound (BD) state are shown. **d** CD (circular dichroism) spectra of the TelG5 without salt (black), in 10 mM KCl (yellow, dashed), and 100 mM KCl (orange). **e** A smFRET histogram of TelG5 with three Gaussian fits (red) of the UF, NP, and P conformational peaks in the presence of 100 mM KCl. The blue-streaked curve represents the sum of the Gaussian fits. **f** Representative time trajectory of the TelG5 exhibiting dynamic transitions between UF, NP, and P conformations in 100 mM KCl. **g** A smFRET histogram of TelG5 incubated with 1 µM of BRCA2OB. The BRCA2OB-bound (BD) state is indicated by the black curve filled with semi-transparent blue. **h** Differential density histogram obtained by subtracting the molecular density of FRET efficiency for TelG5 alone (**e**) from that in the presence of BRCA2OB (**g**). **i** Representative time trajectory of TelG5 in the presence of 1 µM BRCA2OB. The BD state upon binding of BRCA2OB is marked (orange broken line).

if at all, with BRCA2OB, further supporting that BRCA2OB does not bind to UF-ssDNA by itself. The dwell time in the P conformation as well as the NP conformation were shorter in TelG5TTT (NP, 2.3 s; P, 5.2 s) and TelG5TT (NP, 2.1 s; P, 4.3 s), compared to TelG5 (NP, 7.6 s; P, 10.3 s) and TelG5TAA (NP, 6.1 s; P, 11.1 s). This indicates that TelG5TTT and TelG5TT undergo more frequent transitions not only in NP but also in the P conformation, despite their higher P-conformational population (Fig. 3a). Considering that BRCA2OB binding to TelG5TTT and TelG5TT was >9-fold higher in EMSA (Fig. 3c), this marked preference cannot be explained by the shortened dwell times or the gradual (<2 fold) increment in the P-conformational population for TelG5TTT or TelG5TT.

To understand these G4 dynamics further, the transition density between the three interchangeable UF, NP, and P conformations was examined (Fig. 3d). Interestingly, the TelG5TAA and TelG5 substrates mainly displayed transitions between the UF and NP and between the UF and P conformations (Fig. 3d). This indicates that the NP and P conformations structurally rearrange through complete unfolding and refolding in TelG5. In stark contrast, TelG5TTT and TelG5TT predominantly underwent direct transitions between the NP and P conformations (Fig. 3d, e & Supplementary Fig. 9), indicating that the NP-P conformational rearrangement occurs through partially folded intermediate structures without passing through the UF state.

Next, the density of the NP-P direct transition was quantified by measuring the relative fraction of both NP→P and P→NP transitions multiplied by the fraction of molecules undergoing conformational transitions (Supplementary Fig. 10). Notably, the density of the NP-P direct transition matched the relative binding affinity of BRCA2OB to G4 substrates (Fig. 3f). This was true even in the case of TelG5T2G4, in which G4 conformations are more heterogeneous due to the formation of four stacks of G-tetrads (Supplementary Fig. 11). In contrast, addition of BRCA2OB did not alter the characteristic sharp single peak of the smFRET histogram of TelG5CTA [(GGCTAG)₅], indicative of stable G4 folding (Supplementary Fig. 12). This result confirms that BRCA2OB does not associate with TelG5CTA, consistent with the EMSA result (Supplementary Fig. 5). Taken together, these results imply that BRCA2OB does not bind to a stable G4 but selectively interacts with an intermediate topology formed during the dynamic NP-P conformational transition in telomeric G4 (Fig. 3g).

**BRCA2 recognizes and binds to G-triplex-derived intermediates.** Previous studies using molecular dynamics simulations[50–52], stopped-flow kinetic measurement[53,54], NMR spectroscopy[55], and single-molecule experiments[56,57] have proposed the formation of

intermediate structures as part of the G4-folding and unfolding pathway. These intermediates include the G-triplex (G3) and G-hairpin (G2). Taking this information into account, we asked whether BRCA2OB interacts with these intermediates during interconversion between the NP-P conformations. To test this idea, we examined the conformational dynamics of a construct, termed TelG5-G3, where the third GGG triplet within the TelG5 sequence is changed to GTG, resulting in three intact GGG triplets with a long loop of TTAGTGTTA (Supplementary Table 2). The effect of BRCA2OB addition was then tested (Fig. 4a).

In smFRET, the majority of TelG5-G3 alone resided in the low-FRET UF conformation. Notably, a considerable fraction (33% ± 5%, $n = 3$) populated at the two higher-FRET folded states (Fig. 4b, top). The FRET efficiency of the middle-FRET state ($E = 0.58 ± 0.014$; s.d., $n = 3$) was similar to that of G4 conformations (i.e. P and NP). Therefore, this middle-FRET peak is likely to consist of the G3 with the long loop (Fig. 4a) and another type of structure composed of a mixture of G-triads and G-tetrads that may fold through interactions between the G3 and the remaining G bases in the long loop sequence (termed as loop-associated G3:G4, cartoon in Fig. 4b top)[58,59]. Likewise, the high-FRET state ($E = 0.83 ± 0.012$; s.d., $n = 3$) was designated as tail-associated G3:G4 representative, where the G3 is likely to interact with extra G bases in the tail sequence. These structural characterizations were supported by additional experiments with TelG4 [(GGTTAG)₄] and TelG4-G3 [(TTTTAG)-(GGTTAG)₂-(GGTTAT)] (Supplementary Fig. 13). Interestingly, previous reports have identified a molecular structure of G3:G4 and suggested it as an intermediate during G4 conformational transitions[58,59].

Notably, BRCA2OB addition resulted in a rapid growth of the middle-FRET population (G3 and loop-associated G3:G4) and a slight increase in the high-FRET (tail-associated G3:G4) fraction for TelG5-G3, without perturbing their FRET values (middle-FRET, $E = 0.58 ± 0.005$ (s.d., $n = 2$); high-FRET, $E = 0.83 ± 0.009$ (s.d., $n = 2$)) in the presence of BRCA2OB (Fig. 4b, lower). This result indicates that BRCA2OB directly recognizes the G3 and G3:G4 structures. A similar increase in the middle-FRET fraction was observed in TelG5TTT-G3 (Fig. 4c), where the third GGG triplet within the TelG5TTT variant is replaced with GTG (Supplementary Table 2). Note that the FRET efficiency of the middle-FRET state for TelG5-G3 ($E = 0.58 ± 0.005$; s.d., $n = 2$) and TelG5TTT-G3 ($E = 0.58 ± 0.028$; s.d., $n = 3$) perfectly matches to that of the BRCA2OB-bound (BD) state for both TelG5 ($E_{BD} = 0.57 ± 0.006$; s.d., $n = 4$) and TelG5TTT ($E_{BD} = 0.57 ± 0.011$; s.d., $n = 5$) (shaded in Fig. 4d; adopted from Fig. 2g and Supplementary Fig. 7c).

The binding of BRCA2OB to the G-triplex-derived structures (i.e., G3 and G3:G4) was confirmed by EMSA. Compared to

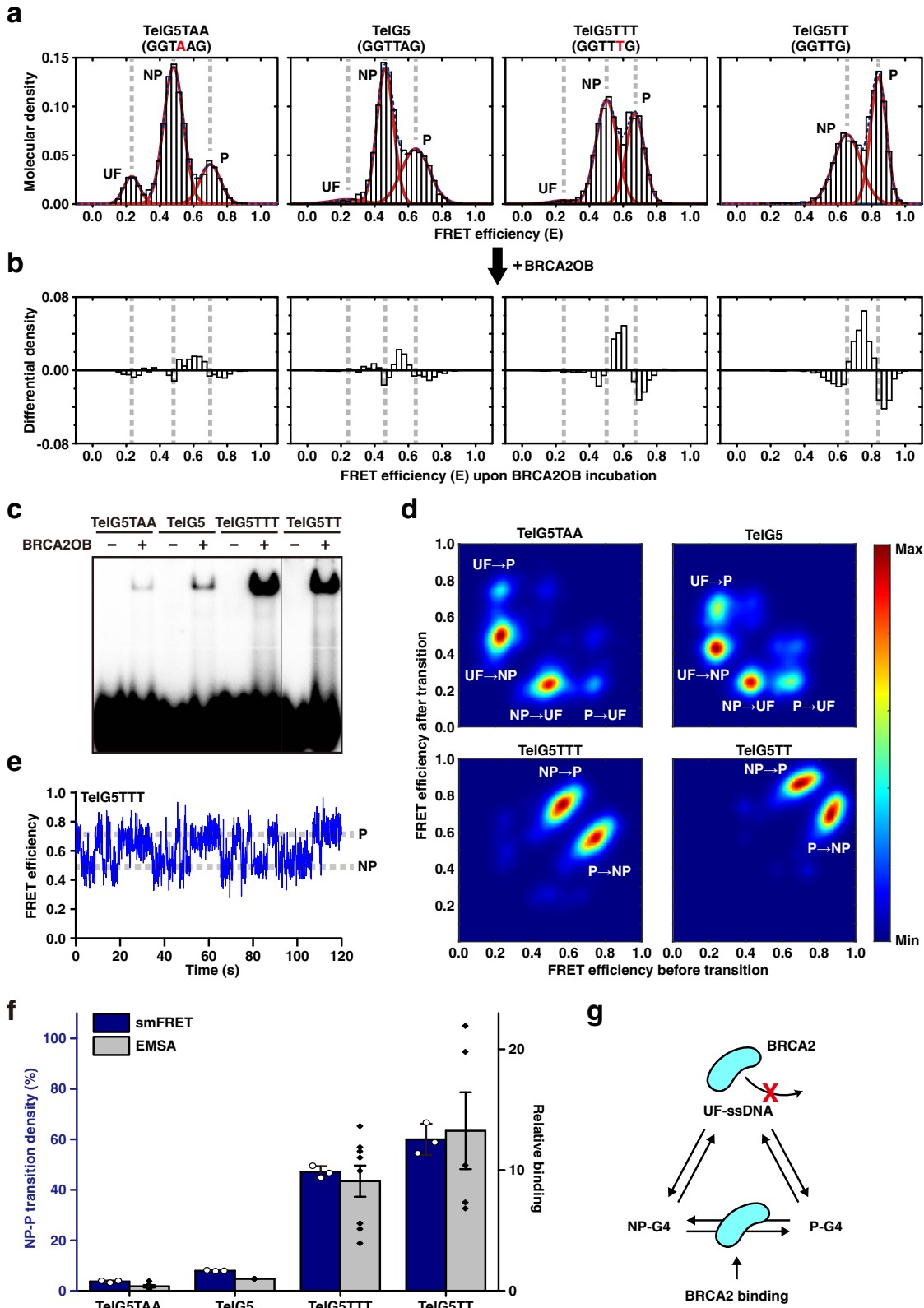

TelG5, BRCA2OB binding was increased by 3-fold for TelG5-G3 and 9-fold for TelG5TTT (Fig. 4e). These results suggest that BRCA2 associates with the G4 substrates primarily through selective interaction with the G-triplex-derived intermediates, which can form during the NP-P conformational interconversions (Fig. 4f, box with broken line). The NP-P conversion via UF-ssDNA (Fig. 4f, lower, unboxed) may also involve the G3-related intermediates. However, BRCA2 bound poorly to G4 substrates that exhibited frequent UF-mediated conformational transitions (Fig. 3f, g): i.e., the half-life of the G3 intermediates in this process, if they exist, is markedly shorter than G3 intermediates formed during the direct NP-P transition. Taken together, BRCA2 recognition and binding to G3 intermediates may reduce folding into G4 (Fig. 4f).

**Fig. 3 Conformational dynamics of the telomeric G4 determines the BRCA2 binding. a** smFRET histograms of the four G4 substrates (TelG5TAA, TelG5, TelG5TTT, and TelG5TT) with Gaussian fits color-coded as in Fig. 2e. **b** Differential density histograms of the four G4 constructs upon incubation with 100 nM BRCA2OB. **c** EMSA results of BRCA2OB binding towards the four G4 substrates. The result **c** is from the same gel, excluding unnecessary lanes. The experiments were repeated at least five times independently. **d** Transition density plots showing relative abundance of each transition (i.e., UF → NP, NP → UF, UF → P, P → UF, NP → P, and P → NP) in the absence of BRCA2OB. **e** A representative time trajectory of the TelG5TTT molecule undergoing direct transitions between the NP and P conformations (without BRCA2OB). **f** A comparative plot between the density of the NP-P direct transition (mean ± s.d., $n = 3$) measured by the smFRET assays (blue bars, left axis) and BRCA2OB binding intensity in EMSA (mean ± s.e.m., $n \geq 5$) in **c** relative to TelG5 (gray bars, right axis). The experiments were repeated independently. **g** A molecular model of the BRCA2-G4 interaction. BRCA2 selectively binds to intermediates during structural rearrangement between the NP and P conformations. All smFRET and EMSA analyses were performed in the presence of 100 mM KCl.

**MRE11, the nuclease that attacks G4 structures in telomeres.** As processivity of replication is slower during lagging strand synthesis, a secondary structure, such as G4, might form from the exposed guanine-rich single strand during telomere lagging strand synthesis. Unresolved G4 can become an obstacle for the progression of the replication fork[9]. The above results indicate a dynamic interaction between BRCA2 and the telomeric G4. In addition, it is known that MRE11 resects stalled replication forks[60]. Therefore, it is reasonable to think that there may be an interplay between BRCA2, telomeric G4, and MRE11. We hypothesized that MRE11 might also be the nuclease that can recognize and resect the telomeric G4 when it becomes an obstacle in fork progression. Interestingly, an older report suggested that yeast Mre11 alone, or as part of a sub-complex, can resect the 5'-end of G-rich single-stranded DNA, the G-tetrad, or the center of TGTG repeats in duplexed DNA[61].

To test whether MRE11 attacks telomeric G4, we purified the nuclease-active form of MRE11 (MRE11N) from *E. coli*[62] (Fig. 5a) and conducted a nuclease assay with the telomere variants. The endo- and exo-nucleolytic activities of recombinant MRE11N were confirmed by its ability to cleave the hairpin-like secondary structured DNA, DAR134 with the nuclease-dead mutant MRE11N-H129N[62] (Supplementary Fig. 14a). Consistently, TelG5GAG, the unstructured single strand, was completely degraded upon incubation with MRE11N, confirming that recombinant MRE11N retained its exo-nuclease activity (Supplementary Fig. 14b).

Upon addition of MRE11N, telomere G4 variants with distinct topological distributions reacted differently (Fig. 5b–e). MRE11 efficiently digested TelG5-G3, TelG5, TelG5TTT, and TelG3GGG in a dose-dependent manner (Fig. 5b–e, lanes 2-3). When 2.5 μM of BRCA2OB was added in nuclease assay towards TelG5-G3, the effect of BRCA2OB was only partial. However, when 5 μM of BRCA2OB was added, smearing bands referring to various sizes of degradation products were apparent with the intact TelG5-G3 band (Fig. 5b, lane 5). These smears are likely to be the products of resected ssDNA, which protruded out of telomere G3 or G3:G4 in TelG5-G3, after the addition of BRCA2OB (Fig. 5b).

Intriguingly, TelG5TTT was most vulnerable to MRE11N in that almost all of the substrate was degraded (Fig. 5d, lane 2-3). To assess whether MRE11 can resect telomeric G4 in a structure-dependent manner, we utilized TelG3GGG, where the TelG5 overhangs were removed resulting in reduced dynamicity[63]. MRE11N also exhibited nuclease activity on TelG3GGG (Fig. 5e, lane 2-3), corroborating that MRE11 can indeed recognize and resect telomeric G4.

Next, we asked whether the addition of BRCA2OB affected MRE11 nuclease activity towards telomere-specific G4. For this, purified BRCA2OB was pre-incubated with the probe before MRE11N addition. The result showed that BRCA2 pre-incubation protected telomeric G4 from MRE11-mediated resection. TelG3GGG was fully protected by BRCA2OB from MRE11 (Fig. 5e, lane 4-5), and 5 μM of BRCA2OB was enough to prevent resection

of TelG5 (Fig. 5c, lane 4-5). TelG5TTT was moderately protected by BRCA2OB in a dose-dependent manner (Fig. 5d, lane 4-5). BRCA2OB preincubation protected the MRE11-mediated degradation in TelG5-G3, but not fully that there were smears (Fig. 5b, lane 5). Smears may be the combination of degraded product of MRE11's nuclease activity towards ssDNA that protruded from BRCA2OB-bound G3. Taken together, these results imply that MRE11 is capable of targeting telomere G4. In addition, BRCA2 binding to the intermediate G3 or G3:G4 conformations (Fig. 4) prevents MRE11-mediated digestion of G4 structures.

**Mutation at the OB-folds of BRCA2 abolishes the association with telomere G4.** Missense mutations at the OB-folds are found in cancer patients, which also exhibit problems in DNA repair[64,65]. Two of these OB-fold mutants were tested for their binding ability to telomere G4. Employing different telomere variants and purified recombinant proteins, EMSA was performed. Both D2723H (mutated in OB1) and R2973C (mutated in OB2) mutants displayed marked decrease in binding to telomere G4s (Fig. 6a). Consistently, binding of wild-type BRCA2OB was lower in TelG5, compared to TelG5TTT, and TelG5T2G4. Nonetheless, mutations at the OB-folds domain led to a marked decrease of binding to all three telomere G4 variants. Note that the binding to G3 intermediates (TelG5-G3) was similarly abrogated by the mutations at the OB-folds domain (Fig. 6a, TelG5-G3).

With compelling lines of evidences, we have shown that the OB-folds domain of BRCA2 can bind to telomere G4 structures. However, whether the full BRCA2 exhibits binding to telomere G4 remained as a question. Therefore, we next asked whether full length BRCA2 exhibits similar binding to telomere G4. Due to the large size (~384 kDa), purification of a recombinant BRCA2 is not feasible. Therefore, we utilized NFLAP-BRCA2-HeLa cells, which stably express EGFP-tagged BRCA2 in a BAC (bacterial artificial chromosome) clone[10] (Supplementary Fig. 1). EGFP-tagged full length BRCA2 was immunoprecipitated with an anti-GFP antibody from NFLAP-BRCA2-HeLa cell lysate. The immunoprecipitate was then incubated with [32]P-labeled probes, followed by several washes and subsequent dot blotting on a positively charged nylon membrane (Fig. 6b). Consistent with BRCA2OB, full length BRCA2 exhibited the strongest binding to TelG5TTT, followed by TelG5T2G4, then TelG5 (Fig. 6c). Next, wild-type *BRCA2-* and full-length OB-folds mutant (*D2723H*; *R2973C*)-expressing constructs, tagged with MBP (Maltose-binding protein), were transfected into 293 T cells and BRCA2 complex were pulled down using amylose resin. Beads were then incubated with [32]P-labeled probes, including TelG5-G3, followed by several washes and subjected to dot blotting as in Fig. 6c. The result confirmed that mutations in OB-folds of BRCA2 abrogated the interaction with telomere G4 and G3-derived intermediates, while wild-type BRCA2 bound to telomere G4 in the order of TelG5TTT > TelG5-G3 > TelG5 (Fig. 6d). Taken together, these

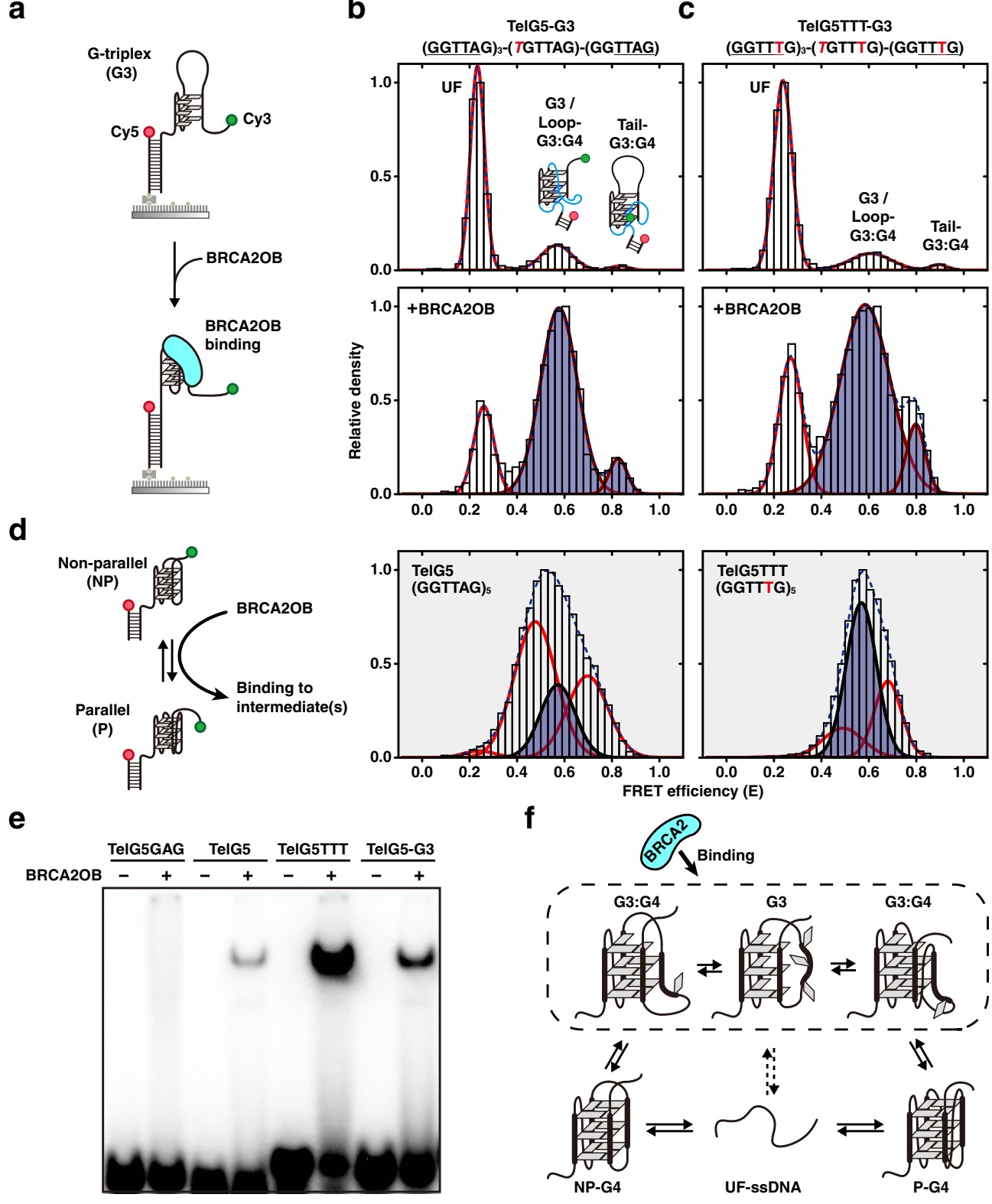

results confirm that the interaction of BRCA2 with telomere G4 is innate to its tumor suppressor activity.

**The mode of BRCA2 and telomeric G4 interaction guarantees telomere replication homeostasis.** Finally, we asked if the interplay between BRCA2, telomere G4, and MRE11 is observed during telomere replication in vivo. In order to test this, we took advantage of conditional *Brca2*-deficient mouse embryonic

fibroblasts (MEFs) immortalized with SV40 LT[66]. In this immortalized mouse fibroblast, which is Telomerase-negative and Brca2-depletion-inducible (TBI), Brca2 can be depleted by the activation of Cre recombinase through the treatment with tamoxifen (4-OHT).

Twenty-four hours post treatment of 4-OHT to deplete Brca2, TBI fibroblasts were transduced with lentiviral-*shMre11* or control (lentiviral-*shGFP*). Forty-eight hours after shRNA introduction,

**Fig. 4 BRCA2 recognizes G-triplex-derived intermediates. a** An experimental scheme to observe BRCA2OB binding towards the dual-labeled TelG5-G3 construct that mimics the G3 intermediate structure. smFRET histograms for **b** TelG5-G3 and **c** TelG5TTT-G3: DNA alone (top) and upon incubation with 1 μM BRCA2OB (bottom) with Gaussian fits (red). BRCA2OB-interacting states are filled in semi-transparent blue. Blue dashed line is the sum of the multiple Gaussian curves. One of the possible topologies of loop-associated G3:G4 (loop-G3:G4) and tail-associated G3:G4 (tail-G3:G4) are illustrated as a cartoon inside the histogram. Flanking tail sequences are underlined for each substrate at the top. **d** A schematic representation (left) of the molecular interaction between BRCA2OB and G4 structures during the direct NP-P transition. For reference, smFRET histograms of TelG5 (middle) and TelG5TTT (right) in the presence of 1 μM BRCA2OB from Fig. 2g and Supplementary Fig. 7, respectively, are shown (filled with gray). **e** EMSA to assess BRCA2OB binding to TelG5-G3, compared to TelG5 and TelG5TTT. TelG5GAG was employed as a negative control. Results are from the same gel. The experiments were repeated three times independently. **f** A model for BRCA2 binding to the G3 and G3:G4 intermediates during G4 conformational interconversions. Two possible G3:G4 conformations and one representative G3 structure that might form during the NP-P rearrangement without complete unfolding are shown. BRCA2 recognizes and binds to G3 intermediates, which reduce folding into G4.

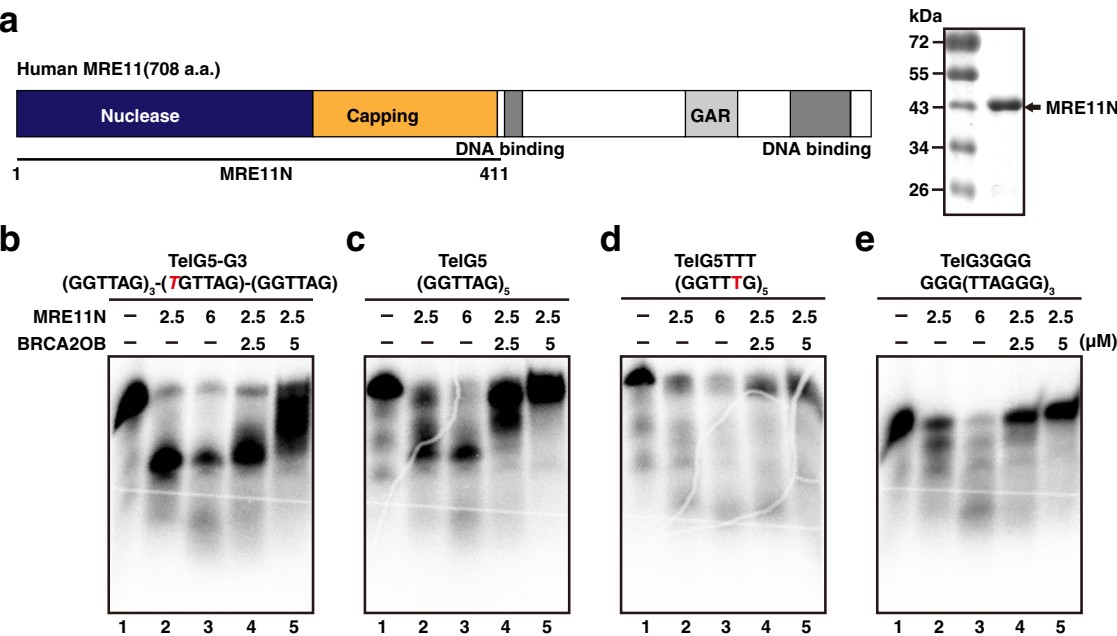

**Fig. 5 MRE11 resects G4 telomere. a** Schematic representation of human MRE11. The functional domains are marked. Recombinant hMRE11 core (1–411 a.a.; MRE11N) was purified, following the previously published method[62] (SDS-PAGE gel on the right). Functionality of the purified protein was assessed (Supplementary Fig. 14) and employed in the nuclease assay. **b, c, d, e** Denaturing PAGE gels showing nuclease activity of MRE11N (lane 2-3) and the effect of BRCA2OB towards various telomere G4 and G3 variants (lane 4-5). 2.5 μM or 6 μM of MRE11N was incubated with indicated DNA substrates: **b** TelG5-G3; **c** TelG5; **d** TelG5TTT; **e** TelG3GGG for nuclease assay. Increasing concentrations of BRCA2OB were employed to assess the inhibition of MRE11-mediated resection. The result is the representative of three independent experiments.

cells were serum starved for 16 hours to arrest the cells in G1/S boundary. Cells were then released into S phase for 4 h with the addition of 16% serum. At this stage, cells were treated with the G4 stabilizer PDS and/or DNA Polα inhibitor Aphidicholin (Aph), which results in replication fork stallings[9], or left untreated. Finally, cells were subjected to immunostaining, using anti-TRF1 and anti-γ-H2AX antibodies to assess telomeric damage (Fig. 7a).

When Brca2 was intact, PDS treatment or the depletion of Mre11 only slightly affected the telomere integrity, as the number of γ-H2AX-positive telomeres per cell was low (Fig. 7b, c, -4-OHT). In contrast, Brca2 depletion increased telomere damage by two-fold (Fig. 7b, c, +4-OHT), which is comparable to the level after Aph treatment in Brca2-intact cells (Fig. 7c, -4-OHT + Aph). Telomere damage in Brca2-depleted cells was further increased to four-fold by PDS treatment (Fig. 7b, c, +4-OHT + PDS). Cells treated with Aph and PDS together exhibited similar level of telomere damage to Aph treated alone, when Brca2 was intact (Fig. 7c). Notably, Aph treatment or PDS treatment exhibited additive effect with the abrogation of Brca2 (Fig. 7c, +4-OHT + Aph; +4-OHT + PDS). Treatment of Aph and PDS in Brca2-depleted TBI cells (Fig. 7c, +4-OHT + Aph

+PDS) displayed similar level of telomere damage as to the single drug treatment (+4-OHT + Aph; +4-OHT + PDS), comparable to the result of Brca2-intact cells treated with Aph and PDS together (-4-OHT + Aph +PDS).

The effect of Mre11 depletion in telomere damage was marginal when Brca2 was intact (Fig. 7b, -4-OHT + shMre11). However, in the absence of Brca2, influence of Mre11 was significant: Mre11 depletion reduced γ-H2AX-positive telomeres generated by the abrogation of Brca2 (Fig. 7b, +4-OHT + shMre11). Mre11 depletion also reduced PDS-induced telomere damage in Brca2-depleted cells (Fig. 7b, +4-OHT + PDS + shMre11) to the level comparable to Brca2 abrogation alone (Fig. 7b, +4-OHT). Similar results were obtained when the MRE11 inhibitor Mirin was applied (Supplementary Fig. 15), corroborating that BRCA2 prevents MRE11-mediated collapse of G4-stalled replication forks.

Brca2 loads Rad51 onto the sites of stalled replication forks in telomere and facilitates telomere replication[8]. Previously, we have reported that abrogation of Brca2 leads to fragile telomeres, signs of collapse of stalled telomere replication forks, and subsequently induces break-induced replication (BIR) and ALT (Alternative

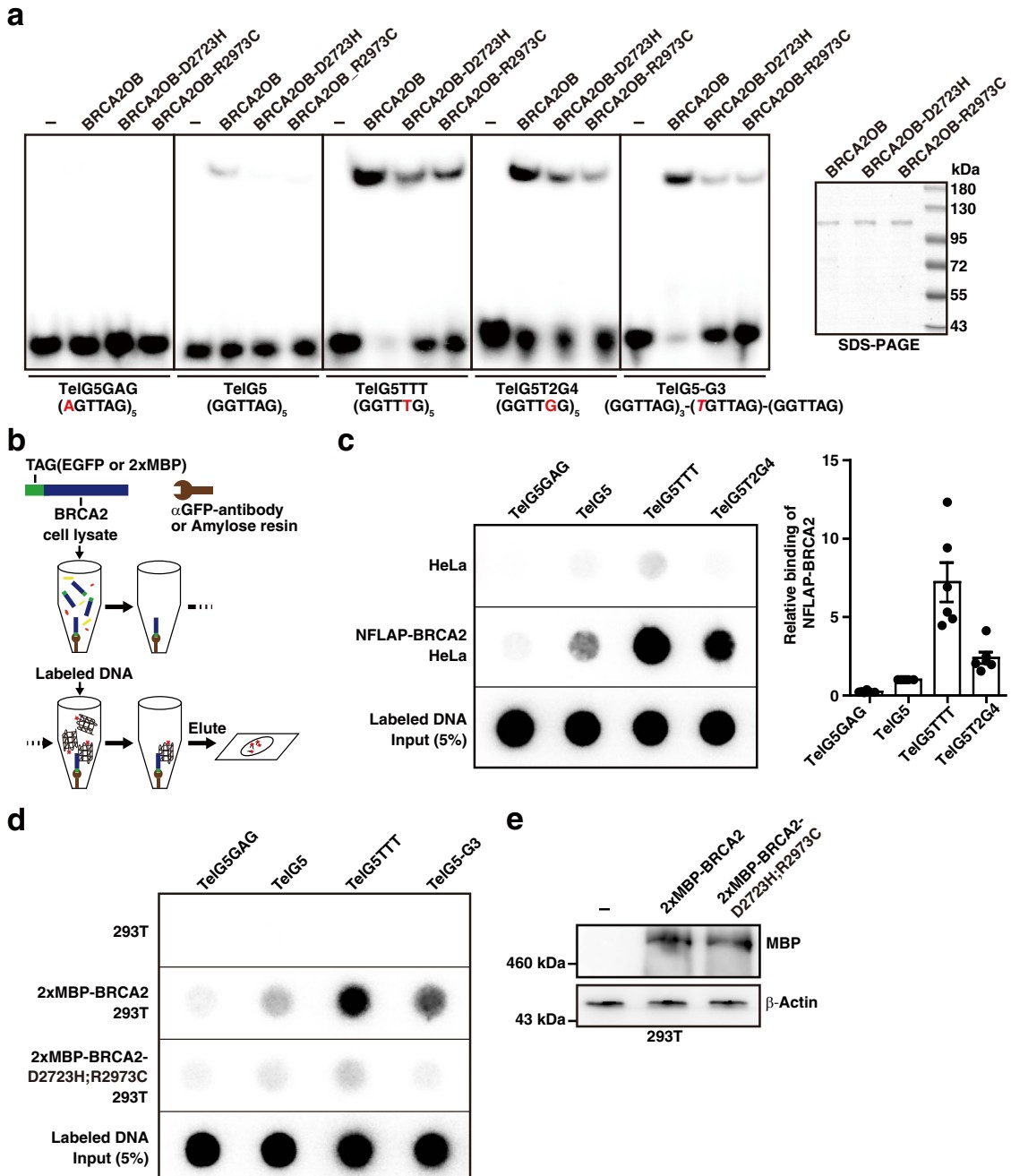

**Fig. 6 Pathogenic mutation in the OB-folds of BRCA2 abolishes the BRCA2 binding to telomeric G4. a** Recombinant BRCA2OB proteins mutated at D2723H and R2973C, respectively, were subjected to EMSA with telomere probes (left). The SDS-page gel of purified BRCA2OB, BRCA2OB-D2723H, BRCA2OB-R2973C mutants (right). **b–d** Binding properties of full-length BRCA2 with indicated telomere variant repeats. **b** Schematic illustration of the assay. EGFP-tagged full-length BRCA2 was immunoprecipitated from NFLAP-BRCA2 HeLa cells[10,78] and incubated with labeled probes in the presence of 100 mM KCl. Anti-GFP antibody was employed to pull down NFLAP-BRCA2 from the cell lysate. Unbound probes were washed out and NFLAP-BRCA2-bound probes were eluted and subsequently dot blotted. **c** Dot blots showing the telomere binding properties of full-length BRCA2. HeLa cell lysate served as negative control. Five percent out of the total labeled probes were loaded as controls. Bar graph indicates the relative binding intensity of full-length BRCA2 to DNA probes (mean ± s.e.m.). The result is from six independent experiments. **d** The effect of OB-folds mutation in binding to telomere G4. Full-length wild-type and mutant *BRCA2*-expression constructs, tagged with 2X MBP (Maltose binding Protein), were transfected into 293 T cells. Forty-eight hours later, 293 T cell lysates were subjected to BRCA2 pull-down using amylose resin, then subjected to binding to various telomere G4 variants, including TelG5-G3, and washed. Bound probes were dot blotted as in **b** and **c**. Five percent of the probes were blotted for control. The result is the representative of three independent experiments. **e** Western blot analysis of wild-type and mutant BRCA2, performed simultaneously with **d** after transfection into 293 T cells to assess the expression level of wild-type vs mutant BRCA2 level. The result was repeated in two independent experiments.

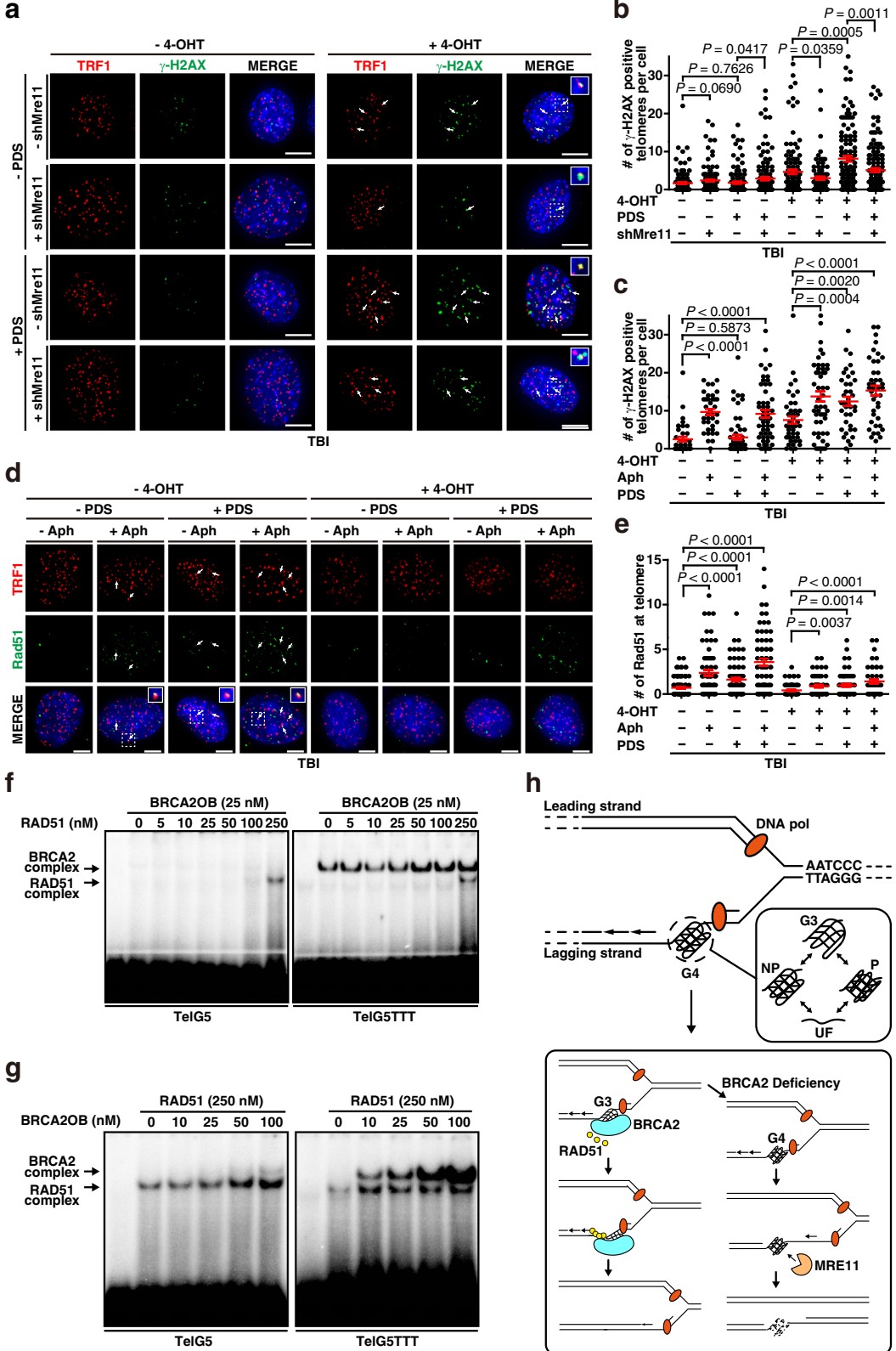

Lengthening of Telomeres). In this case, depletion of Rad51 paralogues in Brca2-abrogated TBI fibroblasts did not alter the ALT activity, indicating that RAD51 paralogues are not required for BIR per se[66]. Nevertheless, as RAD51 is implicated in replication restart and progression under replicative stress[67,68], we asked whether Rad51 localization at the stalled replication sites was affected by the presence or absence of Brca2. TBI

fibroblasts were treated with or without PDS or Aph in the presence and absence of Brca2. Immunofluorescence with anti-TRF1 and -Rad51 antibodies were scored and the Rad51 localization to telomeres were assessed. The result showed that Rad51 localized to telomeres when cells were treated with Aph or PDS in Brca2-intact cells (Fig. 7d, e, -4-OHT), indicating that Rad51 localizes telomere G4-induced stalled forks. However,

**Fig. 7 BRCA2 protects and remodels G4-driven stalled replication forks at telomeres. a** Immunofluorescence images of telomere damage. TBI fibroblasts (telomerase-null (*mTR*⁻/⁻); *Brca2*^F11/F11^; *Cre-ER*^TM^), in which Brca2 depletion can be controlled by the treatment of tamoxifen (4-OHT)[66] (Supplementary Fig. 16), were serum starved for 16 hours then released. PDS was treated at the point of release when needed. Cells were subjected to immunostaining with anti-TRF1 (red) and anti-γ-H2AX antibodies (green), 4 h post release. Scale bar, 5μm. Inset, enlarged foci marked with dotted square. **b** Plots representing the number of damaged telomeres per cell from **a**. Number of cells analyzed (*n*): Control, *n* = 126; +shMre11, *n* = 129; +PDS, *n* = 129; +PDS + shMre11, *n* = 131; +4-OHT, *n* = 121; +4-OHT + shMre11, *n* = 92; +4-OHT + PDS, *n* = 130; +4-OHT + PDS + shMre11, *n* = 137. **c** Plots representing telomere damage after treatments with (+) or without (−) 1.4 μM Aphidicholin (Aph) and/or 5 μM PDS. Drugs were treated for 16 h then subjected to immunofluorescence with antibodies as in **a**. Control, *n* = 39; +Aph, *n* = 37; +PDS, *n* = 61; +Aph +PDS, *n* = 54; +4-OHT, *n* = 45; +4-OHT + Aph, *n* = 43; +4-OHT + PDS, *n* = 38; +4-OHT + Aph +PDS, *n* = 42. **d** Rad51 localization at telomeres after the indicated treatments. Red, immunofluorescence with anti-TRF1; green, anti-Rad51. Scale bar, 5 μm. **e** Rad51-positive telomeres per cell from **d**. Control, *n* = 99; +Aph, *n* = 66; +PDS, *n* = 78; +Aph +PDS, *n* = 78; +4-OHT, *n* = 77; +4-OHT + Aph, *n* = 53; +4-OHT + PDS, *n* = 61; +4-OHT + Aph +PDS, *n* = 45. **f** Result of EMSA with indicated probes when increasing doses of RAD51 was added and co-incubated with 25 nM of BRCA2OB. **g** EMSA result when increasing doses of BRCA2OB was added and co-incubated with RAD51. The result is the representative of three independent experiments (**f**, **g**). **h** Model for how BRCA2, RAD51, and MRE11 work with telomeric G4 during replication.

absence of Brca2 compensated the effect from Aph and/or PDS treatment for Rad51 loading at telomeres (Fig. 7d, e, +4-OHT). These results speak to the fact that BRCA2 is essential for loading RAD51 to the sites of G4-driven stalled replication forks at telomeres.

If BRCA2 binds to telomere G4 by recognizing G3 intermediates to facilitate replication fork progression at telomeres, how would RAD51 work at the telomere G4? To answer to this question, BRCA2OB and recombinant RAD51 together were employed in EMSA. TelG5 and TelG5TTT were employed in the assay to represent and compare the dynamic nature of telomere G4. The result confirmed that BRCA2OB binds better to TelG5TTT, the construct that represents the interconversion between NP- and P-telomere G4, much better than TelG5, which mainly undergoes unfolded ssDNA-mediated structural dynamics with some NP-P direct transitions (Fig. 7f). Meanwhile, RAD51 binding to telomere was not as efficient that only when high concentration of RAD51 (250 nM) was included, RAD51 complex was observed both in TelG5 and TelG5TTT (Fig. 7f, RAD51 complex). Note that the bands corresponding to BRCA2 and RAD51 complex do not compete for each other.

Next, RAD51 was co-incubated with increasing doses of BRCA2OB, and the binding pattern was assessed and compared in TelG5 and TelG5TTT. Interestingly, RAD51 complex bound to the radiolabeled telomere probes increased with the increasing concentration of BRCA2OB in TelG5 (Fig. 7g, left). In TelG5TTT, the formation of RAD51 binding was facilitated by the presence of BRCA2OB. RAD51-TelG5TTT complex remained more or less constant while BRCA2OB-TelG5TTT complex increased in a dose-dependent manner (Fig. 7g, right). These results altogether suggest that BRCA2 recognizes and binds to dynamic telomere G4, whereas RAD51 alone does not. As BRCA2 prefers the binding to G3 or G3:G4 intermediates, the presence of BRCA2 will reduce the level of stable G4. Simultaneously, BRCA2 binding to G3 intermediates will generate unstructured ssDNA, where RAD51 recruited by BRCA2, can bind. This way, BRCA2 protects telomere from the attack of MRE11. Taken together, the mode of BRCA2 association with telomere G4 leads to the remodeling of G4 and facilitates RAD51 to involve in the restart of the stalled replication fork, at the same time preventing the stalled forks from breaking down (Fig. 7h).

## Discussion
In this study, we have provided compelling lines of evidence that BRCA2 interacts with dynamic telomeric G4 through selective binding to G3-derived intermediates, which form during interconversion between parallel and non-parallel telomeric G4 conformations (Fig. 7h, upper). As formation of G3-derived intermediates exposes ssDNA, RAD51 recruited by BRCA2 can

be loaded on to these telomere ssDNA, thereby assisting the restart of stalled replication forks (Fig. 7h, lower left). Furthermore, we identified MRE11 as the nuclease that can attack telomeric G4. Association of BRCA2 with G3 reduces G4, which can be targeted by MRE11. Protection of G3 will be further assisted by the loading of RAD51 onto single-stranded DNA by BRCA2. This way, the interaction of BRCA2 with dynamic G4 structures at telomeres leads to the protection of telomeres from MRE11-mediated resection during replication.

We speculate that BRCA2 interaction with telomeric G4 evolved to maintain genome integrity. It is noteworthy that guanine-rich telomere repeats have been conserved over 1.3 billion years of eukaryotic evolution. Conservation of guanine-richness at telomeres, and folding into G4 structures, may inhibit the access of illicit DNA recombination machinery or even telomerase. Consistent with this hypothesis, telomeric G4 is known to inhibit DNA damage regulators to act at telomeres[37,69,70]. Structural dynamics of telomeric G4 also affects the accessibility of RAD51, WRN, and BLM to telomeres[63]. Notably, the G4-forming property of telomere overhang enhances POT1/TPP1 binding and prevents from DNA damage-inducing RPA to bind to telomeres[37]. Furthermore, POT1/TPP1 first captures telomere G4 but increasing amount of POT1 destabilizes G4[71], facilitating telomere loop (T-loop) formation. The differential binding mode of BRCA2 and POT1 to telomeres (Fig. 2b) suggest that BRCA2 functions at telomere G4 exclusively during replication, but less likely to be found at the T-loop. In this line of thinking, the reason why G-richness is conserved throughout eukaryotic evolution can be explained. First, G4-forming property prevents telomere recombination through capturing POT1/TPP1. Secondly, by adopting dynamic nature compared to G4 found at the promoter region, e.g. *c-MYC* promoter, and the evolution of BRCA2 to associate with the dynamic telomere G4, the mission of maintaining telomere sequence and identity could be achieved. Our results unravel the long-standing question of how specific telomere sequences are linked to telomere function and contribute to genome integrity. Importantly, tumor suppressor BRCA2 plays a pivotal role.

During lagging-strand telomere synthesis, relatively slow synthesis of Okazaki fragments leaves the G-rich telomere template exposed, resulting in their folding into G4. BRCA2 binds to G3-derived intermediates during interaction with telomeric G4, while MRE11 recognizes and attacks. These results provide direct evidence that G4 dynamics can be exploited for specific molecular recognition. We propose that the biophysical nature of dynamic G4 controls accessibility: telomere G4 dynamics may function as a gatekeeper that blocks unwanted interactions, while allowing access to factors that are required for telomere integrity.

Although further structural characterization is required to uncover a detailed mechanism, our in vitro data show that MRE11

exhibits strong nuclease activity towards the G4 substrates. Absence of BRCA2 results in telomere damage in cells, which is further exacerbated upon G4 stabilizer PDS treatment. This observation is in agreement with the previous report by Zimmer and colleagues, which showed that PDS treatment increased telomere damage and reduced the viability of cells lacking BRCA2. These results altogether imply that G4-stabilizing drugs have strong therapeutic potential in BRCA2-deficient tumors[19].

Recently, we showed that Brca2 deficiency results in induction of alternative lengthening of telomeres (ALT) through the instigation of the break-induced replication (BIR)[66,72]. According to the current study, the absence of BRCA2 will increase the level of G4 at telomeres. Therefore, it will be interesting to see whether the increase in telomere G4 is directly associated with the induction of BIR when BRCA2 is dysfunctional. Whether the unique mode of BRCA2-telomere binding is associated with the prevention of ALT should be studied in the future.

Notably, mutations at the OB-folds domain of BRCA2, which are found from human cancers, abrogated the association with telomere G4, indicating that the failure to associate with telomere G4 is pathogenic. The two mutants used in this study, D2723H and R2973C, are located at OB1 and OB2, respectively. A report has shown that D2723H affects a key Asp residue in OB1, which is involved in hydrogen bonding with other residues in OB1 to stabilize helical domain (HD)/OB1 interface, hence the mutant D2723H exhibits high Gibbs free energy, implying destabilization of the protein folding. Notably, D2723H mutation results in the disruption of protein conformation[73]. The D2723H mutation also interferes with DSS1 binding, unmasks the nuclear export signal (NES), hence renders the protein cytosolic[74]. As our experiments were on direct binding assay and nuclease assay with telomere variants (Fig. 5 and 6), the defective interaction with telomere G4 is likely to result from the structural alteration rather than from defective localization of the protein. The fact that we found B2-8, which contains OB2 and OB3 but lacks OB1, associated with telomere G4 (Supplementary Fig. 2) supports this notion. Nevertheless, whether DSS1 is recruited to the ssDNA region of G3 intermediates when BRCA2 binds remains to be elucidated. In comparison, R2973C is known to be less pathogenic and the information regarding R2973C is limited. It is possible that the charged Arg substituted to Cys interferes with the association with DNA. When we utilized the structure prediction software from RoseTTAFold[75], D2723H and R2973C both displayed conformational change, particularly at the tower domain, suggesting that the two mutations at the OB fold interrupt with the structure that associates with telomere G4. Revealing the structure of BRCA2-telomere G4 complex will answer to how these two mutations affect BRCA2 binding to telomere G4. In sum, our data suggest that BRCA2 maintains telomere replication homeostasis through binding to dynamic telomeric G4 structures and remodels it thereby facilitating the restart of the stalled fork. The unique mode of BRCA2 interaction with telomere G4 is essential for its tumor suppressor activity.

## Methods

**Statistical analysis**. Graphpad Prism V5 (GraphPad Software Inc., San Diego, CA) was used for statistical analyses. Two-sided Student's t-test was used unless otherwise stated. Mean ± s.e.m. is shown wherever required unless described otherwise.

**Cloning, expression and purification of recombinant proteins**. BRCA2OB (2665–3197 a.a.) and POT1OB (10–278 a.a.) were inserted into pMAL-c5X (New England BioLabs) with a 6 x His-tag added at the C-terminus. They were expressed in E. coli and induced using 0.1 mM IPTG at 18 °C overnight. Cells were lysed in lysis buffer (50 mM Tris-HCl pH 7.5, 150 mM NaCl, 2 mM β-mercaptoethanol, 3 mM MgCl$_2$, 10 mM imidazole, 0.5% NP-40, 1 mM EDTA, and 1 mM PMSF), followed by sonication. The lysate was centrifuged at 50,000 G for 1 h, and the

proteins were eluted using Ni-NTA agarose (QIAGEN). The mutants of BRCA2OB were prepared based on pathogenic mutations by site-directed mutagenesis and the recombinant proteins were purified. MRE11N (1–411 a.a.) was inserted into pET-28a (Novagen) and MRE11N_H129N was obtained by site-directed mutagenesis, and the recombinant proteins were purified as described. Human RAD51 was purchased from Sigma–Aldrich (recombinant, expressed in E. coli, SRP2090).

**Antibodies and cell lines**. Antibody to mouse Brca2 was described previously[9,11,66]. The following antibodies were purchased and used in the indicated dilution ratio: anti-MBP mAb (E8032, New England BioLabs, 1:10,000); anti-β-actin rabbit mAb (A700-057, Bethyl Laboratories, 1:1000); anti-Mre11 rabbit pAb (4895 S, Cell signaling Technology, 1:1000); anti-gamma H2AX (ser139) rabbit mAb (9718 S, Cell Signaling Technology, 1:200); anti-TRF1 mouse mAb (TRF-78, ab10579, Abcam, 1:200); and anti-Rad51 rabbit pAb (Ab-1) (PC130, Calbiochem, 1:500).

TBI fibroblasts (telomerase-null ($mTR^{-/-}$); $Brca2^{F11/F11}$; Cre-ER$^{TM}$) was described previously[66]. NFLAP-BRCA2-HeLa cell line was provided from Mark Petronczki[10]. Human 293 T cells (CRL-3216) were purchased from ATCC.

**Electrophoretic mobility shift assay (EMSA)**. 10 nM γ-$^{32}$P-end-labeled DNA probes were pre-incubated at 37 °C for 15 min in EMSA buffer (50 mM Tris-HCl pH 7.5, 100 mM KCl, 2 mM β-mercaptoethanol, and 10 % glycerol) and the purified protein was added. When RAD51 was incubated, 1 mM MgCl$_2$ and 2 mM ATP were added into the EMSA buffer. After incubation, the mixture was loaded onto a 5% polyacrylamide gel containing 0.5 X TBE buffer, and the gels were run at 10 V cm$^{-1}$ for 2 hr at 4 °C. Gels were dried using a 583 Gel dryer (Bio-rad) and data was captured using the Fuji FLA 7000 scanner. The signal intensity was quantified using Multi Gauge V3.0 software (Fujifilm, Tokyo, Japan). Data were analyzed by curve fitting and regression (Prism V5; GraphPad Software Inc., San Diego, CA) to determine the equilibrium dissociation constant, using the one site specific binding equation: $Y = Bmax*X/(K_d + X)$ (Bmax = 1 set as a constraint). Sequences of DNA probes used in EMSA are listed in Supplementary Table 1.

**Circular dichroism (CD) spectroscopy**. To measure the CD spectra, 10 μM DNA probes were prepared by heating at 95 °C for 10 min and cooling down to room temperature in 0~100 mM KCl, 50 mM Tris-HCl pH 7.5. The CD spectra were recorded on JASCO J-815 spectropolarimeter between 220–320 nm of wavelength using 1 mm path length quartz cuvette at 37 °C. Each CD spectroscopy measurement is the average of three scans with 200 nm min$^{-1}$ scanning speed. The sequences of DNA probes in CD are listed in Supplementary Table 1.

**Single-molecule FRET DNA sample preparation**. DNA oligonucleotides for single-molecule experiments were purchased from Integrated DNA Technologies (IDT). The G4-forming DNA strands were amino-modified at the 3' end, while a short complementary strand for partial duplex (stem) formation was 5' amino-modified and 3' biotinylated for surface immobilization. Fluorophores (Cy3 or Cy5) were coupled to DNA using NHS-ester conjugated dyes (GE Healthcare). G4 and partial duplex formation were induced by slowly annealing the DNA substrates from 95 °C to room temperature in 250 mM KCl, 25 mM Tris-HCl pH 8.0. The sequences of DNA constructs, including biotin and/or amino modifications, are listed in Supplementary Table 2.

**Single-molecule FRET measurement and data analysis**. Single-molecule FRET experiments were performed using a home-built prism-type total internal reflection fluorescence microscope[76]. A single-molecule imaging chamber was assembled with a microscope quartz slide and glass coverslips that were coated with poly-ethylene glycol and surface-biotinylated. For all single-molecule imaging, biotinylated DNA constructs were immobilized on the chamber surface via the biotin-NeutrAvidin interaction and imaged with or without BRCA2OB at 37 °C in imaging buffer [50 mM Tris-HCl pH 7.9, 100 mM KCl, 20 mM NaCl, 1 mM DTT, 0.1 mg ml$^{-1}$ BSA, and 9% (v/v) glycerol] with an oxygen scavenging system [1 mg ml$^{-1}$ glucose oxidase (Sigma-Aldrich), 0.04 mg ml$^{-1}$ catalase (Sigma-Aldrich), and 0.8% (w/v) β-D-glucose (Sigma-Aldrich)] and a triplet quencher [~3 mM Trolox (Acros Organics)] for enhanced photostability. Fluorescence signals from individual DNA molecules were recorded with a time window of 2 s and 200 s (100 ms of the acquisition time) in order to construct smFRET histograms and time trajectories, respectively, using an open-source program Single (downloaded from https://github.com/pjb7687/single/). In the absence of BRCA2OB, images were collected for ~20 min intermittently (after 30 min of pre-incubation of the constructs at 37 °C in the case of TelG5-G3 and TelG5TTT-G3), while the FRET histograms and time trajectories in the presence of BRCA2OB were obtained 5 min and 10 min after protein addition, respectively. For FRET histograms, green-red alternating laser excitation was introduced when necessary during data acquisition in order to focus on dual-labeled molecules. Data from over 2000 molecules were used to build each histogram, and fitted by multiple Gaussian curves; for histograms in the presence of BRCA2OB, the center FRET values of the UF, NP, and P conformational peaks were fixed to those obtained without the protein (or restricted to the protein-free values ± 0.01). The relative population of each FRET state was quantified as the relative area under each Gaussian fit curve.

For the transition-density and dwell-time analyses, individual time trajectories were categorized manually into docked or transitioning molecules with the criterion of whether the trajectory exhibits FRET transitions or not during the first 100-s data points. Then, fragments of the transitioning trajectories (from >200 molecules) were collected, and were fitted into 2-4 states using open-source software vbFRET[77] to construct the transition density and dwell time plots. Data processing and analysis was performed using home-written or open-source IDL and MATLAB codes, which can be downloaded from https://github.com/Ha-SingleMoleculeLab/Raw-Data-Analysis/, and Igor Pro.

**Nuclease assay**. 10 nM or 1 nM of γ-$^{32}$P-end-labeled DNA substrate was incubated in reaction buffer [25 mM MOPS (pH 7.0), 100 mM KCl, 1 mM MnCl$_2$, and 1 mM DTT] for 15 min at room temperature prior to reaction. Approximately 2.5 or 6 μM of purified MRE11N was incubated at 37 °C for 60 min. Reactions with BRCA2OB and MRE11N were identical except 2.5 μM or 5 μM of BRCA2OB was pre-incubated at 37 °C for 15 min prior to addition of MRE11N. The reaction was stopped by adding 1/10 volume of stop mixture (2% SDS, 100 mM EDTA, and 0.5 mg ml$^{-1}$ proteinase K), followed by a 5 min incubation at 37 °C. Reaction products were resolved on 15% denaturing polyacrylamide gels containing 7 M Urea in 1 X TBE buffer. Gels were run for 120 min at constant power and were fixed with buffer (30% methanol, 5% acetic acid, 5% glycerol). The fixed gels were dried and imaged.

**Dot blotting to assess full-length BRCA2 binding to telomere variants**. NFLAP-BRCA2-HeLa cells and HeLa cells were lysed in IP buffer [50 mM Tris-HCl pH 7.5, 10% glycerol, 150 mM NaCl, 1 mM DTT, 1 mM EDTA, 0.2% Triton X-100, 5 mM NaF, 200 μM Na-orthovanadate and protease inhibitor cocktail (Roche)][78], and sonication was performed. The lysates were incubated with 1 mM MgCl$_2$ and benzonase for 1 h at 4 °C and centrifuged at 15,000 G for 10 min. The supernatant was transferred to GFP-Trap Agarose bead (ChromoTek) and incubated overnight at 4 °C. After the wash, beads were incubated in IP buffer containing 15 mM MgCl$_2$ and 5 mM ATP, followed by incubation with γ-$^{32}$P-end-labeled DNA probes in EMSA buffer. BRCA2 bound probes were eluted from the bead by boiling at 95 °C, while unbound probes were washed out. The eluted probes were dot-blotted onto a positively charged nylon membrane (Amersham™ Hybond™-N +, cytiva) and imaged as previously described. In case when wild-type and mutant BRCA2 were assessed and compared for binding to telomere G4 variants, MBP (Maltose binding protein)-tagged wild-type and mutant BRCA2-expressing constructs were transfected into 293 T cells. Forty-eight hours post transfection, cell lysates were prepared and subjected to binding to amylose resin (New England BioLabs) to pull down MBP-tagged BRCA2. The BRCA2-bound beads were incubated with radiolabeled probes in EMSA buffer, followed by several washes. The eluates were then dot blotted as above.

**Immunofluorescence assay**. MEF cells were cultured with or without 1 μM of 4-OHT (Sigma-Aldrich) for 24 h and treated with lentiviral shMre11 for 48 h. To generate the lentiviral *shMre11*, psPAX2 (#12260, Addgene), pMD2.G (#12259, Addgene), and pLKO.1 (#8453, Addgene, subcloned shMre11 oligonucleotide) were transfected into 293 T using polyethylenimine. Lentiviral infection to target cells was performed in the presence of 4 μg ml$^{-1}$ of polybrene (Hexadimethrine bromide, H9268; Sigma-Aldrich). Sequence of *shMre11* is as follows: sense, 5'-TACAGGAGAAGAGATCAACTTTCAAGAGAAGTTGATCTCTTCTCCTGTTTTTTTTC-3', antisense, 5'-TCGAGAAAAAAACAGGAGAAGAGATCAACTTCTCTTGAAAGTTGATCTCTTCTCCTGTA-3'. MEFs were serum starved for 16 h after infection, then cultured in the absence or presence of 5 μM pyridostatin (PDS, Sigma-Aldrich) for 4 h. The cells were grown on a coverslip and fixed with 4% paraformaldehyde, permeabilized in PBS containing 0.5% Triton X-100 (0.5% PBS-T) for 15 min at room temperature. The coverslips were blocked in 10% goat serum in 0.1% PBS-T for 2 h at room temperature followed by overnight incubation with primary antibodies in blocking solution at 4 °C. The cells were incubated with secondary antibodies for 2 h at room temperature and mounted with VECTA-SHIELD mounting medium containing DAPI (Vector Laboratories). Microscopic images were acquired using DeltaVision Cell imaging system (PersonalDV, Applied Precision/GE Healthcare, USA).

**Reporting summary**. Further information on research design is available in the Nature Research Reporting Summary linked to this article.

## Data availability
The data supporting the findings of this study are available from the corresponding authors upon reasonable request. Source data for the figures and supplementary figures are provided as a Source Data file. Source data are provided with this paper.

## Code availability
The open-source programs and codes used for smFRET data acquisition and analysis can be downloaded from https://github.com/pjb7687/single/, https://github.com/Ha-SingleMoleculeLab/Raw-Data-Analysis/, and http://vbfret.sourceforge.net/. All other auxiliary custom codes are available upon request to the corresponding authors.

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

## Acknowledgements

We thank the members of Hyunsook Lee's lab and Seong Keun Kim's lab for technical assistance and critical discussions throughout the study. Special thanks to Jennifer J. Lee for help with immunofluorescence experiments, Si-Young Choi for running RoseTTAFold for structure prediction and discussion, and Jinho Park for discussion on smFRET results. We are grateful to Sabi Lall and Aiden H. S. Yoo for editing this manuscript. This work was supported by National Research Foundation of Korea (NRF) (2020R1A5A1018081 and 2021R1A2C1006191) to H.L. This work also was supported by

the National Research Foundation of Korea (NRF) (NRF-2018R1A2B2001422) to S.K.K. J.Y.L is a postdoctoral fellow of the Brain Korea 21 FOUR program from National Research Foundation (NRF), Korea (4199990314450) and K.S. was supported by the Global PhD Fellowship from the NRF (NRF-2018H1A2A1060095).

## Author contributions

H.L. and S.K.K conceived, supervised, and led the study. H.L. took the main role in writing the manuscript. J.L. designed and performed overall biochemical and cellular experiments. K.S. designed and performed single-molecule assays and interpreted the data. S.Y.J. conducted MRE11 nuclease assays and BRCA2OB mutant studies. J.-H.J. revealed the region of BRCA2 that binds to the telomere G4. The manuscript was written and approved by all authors.

## Competing interests

The authors declare no competing interests.
