## [Peer Review File · Nature Communications]

Title: Dynamic interaction of BRCA2 with telomeric G-quadruplexes underlies telomere replication homeostasisReviewers' comments:

Reviewer #1 (Remarks to the Author):

Dynamic interaction of BRCA2 with telomeric G-quadruplexes underlies telomere replication homeostasis

The authors studied the molecular mechanism underlying the replication fork collapse during lagging-strand telomere synthesis in which G-quadruplex (G4) structure is expected to form and BRCA2 may interact with the G4 structure. Using gel shift assay combined with single molecule FRET method, the authors demonstrate that BRCA2 associates with G-triplex (G3)-derived intermediates which is thought to be involved in the interconversion between parallel and non-parallel G4 structures. They proposed a model where BRCA2 target such intermediate structure. They show that MRE11 nuclease G4 resecting activity is inhibited in presence of BRCA2. Using mouse strains, they also provide evidence that BRCA2 depletion results in increased telomeric damage. The study suffers from not having a clear premise, inconsistent data interpretation, poor data quality and incoherent narrative. This manuscript cannot be published in its current form.

Major points:

1. Many published studies clearly indicate that BRCA2 functions with DSS1 in associating with single stranded (not structured) DNA for the purpose of recruiting RAD51 for HR activation. Here are the supporting publications.

- Yang et al (Science 2002) resolved a co-crystal structure of BRCA2-DBD with DSS1 bound to oligo (dT)₉ single strand DNA which clearly revealed the specific contact of OB1 and 2 domains of BRCA2-DBD with single stranded DNA bases (5-6 nucleotide) in the presence of DSS1. BRCA2-DBD-DSS1 complex, not BRCA2-DBD alone represents the cellular unit that binds single strand DNA.

- Zou et al (Oncogene, 2006) demonstrated that DSS1 is required for the stability of BRCA2. In this study, depletion of DSS1, similarly to the depletion of BRCA2, resulted in hypersensitivity to DNA damage, again indicating that BRCA2 does not act alone in associating with DNA.

- Zhao et al (Mol Cell, 2016) showed a functional significance of BRCA2-DSS1 in RPA-RAD51 mediated homologous recombination.

- Le et al (NAR, 2020) further verified the functional unit of BRCA2-DSS1 in HR and also showed that DSS1 and single strand DNA regulate BRCA2 oligomerization.

Based on the previous studies, the fragment of BRCA2 the authors studied with structured (telomeric) DNA is likely biologically irrelevant.

2. The widely known function of BRCA2 in DNA damage and telomere damage is the homologous

recombination in which BRCA2 works in conjunction with RAD51. The telomere shortening that occurs in BRCA2 deletion can also be detected in RAD51 deletion, supporting the role of both proteins in HR. Therefore, if BRCA2 association with telomeric G4 represents a functional state, it should be tested by its ability to recruit RAD51 and stimulate HR or the D-loop formation.

3. Related to above, since BRCA2's known role is to recruit RAD51 for HR, the conclusion from the current study, especially the simple binding (without stretching or unfolding) of BRCA2 on telomere does not make sense in terms of the known activity of BRCA2 in HR pathway.
4. The binding affinity to telomere substrate (TelG5) is negligible, making it impossible to measure the binding constant. The higher affinity of BRCA2-DBD binding to non-telomeric G4 reveals that telomeric G4 is not a relevant substrate for BRCA2. The interaction is likely nonspecific.
5. The CD result and P/NP designation is counterintuitive and inconsistent with previous results from other laboratories.
6. Single molecule data is of poor quality and the histogram, density plot etc. are over-fitted by selecting center positions. As a result, the assignment of BD state is not valid in many cases.
7. There is no experimental evidence that BD occurs in transition between the two G4 folded states. The interconversion model depicted in Figure 5f is incorrect.
8. The MRE11 activity test is of data quality that is not acceptable for publication.

Figure by figure

1. In Figure 1, the EMSA gel images display that extremely small fraction of binding while majority of the labeled DNA remains unbound (thick bottom staining) in all conditions tested, suggesting that this is not specific binding. G-quadruplex, due to the folded nature of the nucleic acid has high propensity to nonspecifically interact with various proteins (Begeman, EMBO reports, 2020). By contrast, DNA binding proteins such as RPA and POT1 exhibit nanomolar affinity and POT1 displays high specific binding to telomeric single strand DNA.
2. How do the authors determine the dissociation constant (K_d) of BRCA2 in different G4 constructs when the K_d values must be higher than the binding fraction used in the graph? If measured K_d of BRCA2 to TelG5 is 4.4 μ M then why use 1 μ M of protein concentration during their measurements.
3. If the authors seek to study BRCA2-DBD binding affinity comprehensively, they should test unstructured DNA such as T8-T36 used in the Yang et al's (Science 2002) structural study so that the BRCA2-DBD can be directly compared with BRCA2-DBD-DSS1.

4. In Figure 2, the binding constant is tested by EMSA. Again, due to the low binding affinity, the K_d cannot be determined accurately. Most strikingly, the telomeric substrate, TelG5 exhibits the lowest binding affinity while the non-telomeric substrates (TelG5 variants) show an improved binding. It is clear from this result that the low level of BRCA2 binding to TelG5 does not represent functional state of BRCA2 at telomere. Although the authors refer to sequences such as TTTGGG as telomere variant repeats, it is highly unlikely to encounter (GGTTTG)₄ or higher repeat number in human telomeres.

5. Figure 3b top, the CD changes in low vs. high KCl is opposite of what others have shown. The parallel (260nm peak) should be stabilized by high potassium concentration. In addition, the authors mentioned ~48% molecules showed dynamics between NP-P transition of TelG5, but the dynamics may arise from the flanking sequences outside of the folded G4.

6. Based on the 260 vs. 290nm distribution in CD measurement, the authors assigned FRET histogram peaks as parallel and non-parallel which is not entirely consistent with the dye positions on the DNA. The single stranded regions on both ends of the G-quadruplex likely make this interpretation difficult, also results in highly noisy fluorescence signals in individual traces.

7. The fitting of BD state located within the NP and P states is barely possible. It's not clear what fraction of individual traces exhibit distinguishable transitions.

8. Author used the TDP to show the transition between the states of UF, NP, and P. TDP for TelG5TAA and TelG5 is not reported. There is a discrepancy as the histogram doesn't show any unfolded population for TelG5 but the TDP showed the major transition from unfolded state. Author should clarify this discrimination.

9. Author mentioned that "Notably, the density of the NP-P direct transition matched the relative binding affinity of BRCA2OB to G4 substrates (Fig. 4f)." However, the relative BRCA2 binding is only ~10-20% from EMSA whereas ~60% same binding from smFRET measurement for TelG5TTT and TelG5TT. This is a

10. Figure 4, the BD state assignment, again is not clear, especially for TelG5TAA and TelG5. Figure 4e shows TelG5TTT trace, yet no BD state is shown.

11. There is no clear evidence that BRCA2 binds to intermediate between NP and P. If the traces show clear transition between P, NP and BD, then the authors can analyze the partition between P to BD vs. NP to BD.

12. Authors propose a model for NP-P transition during which BRCA2 engages. The propose model of NP-G4 to P-G4 transition at Fig. 5f where the fourth strand remains in the same orientation i.e

unchanged. According to the FRET peak, the fourth strand must be oriented differently to induce a different FRET state. This model is misleading.

13. In Figure 5, the G-triplex diagram does not match the FRET positions. The high fraction of low FRET indicates mostly unfolded state of the DNA. Again, the binding is not significant as shown on the gel.

Reviewer #2 (Remarks to the Author):

In this study, the authors provide compelling biochemical evidence for the ability of BRCA2 to interact with G4-forming telomeric DNA and perturb its structure, most likely by binding to folding intermediates between different conformations of G4. They also demonstrate that Mre11 is able to cleave DNA containing G4, and this ability is blocked by BRCA2. The experimental data are robust, and include complementary lines of evidence from both EMSA and single molecule FRET analyses. The FRET analyses, in particular, allow the authors to probe the molecular interactions between BRCA2 and G4 in some detail. Finally, a small amount of in vivo analysis in BRCA2-deficient mouse cells supports a synergistic effect of BRCA2 loss and G4 stabilization on telomeric DNA damage, which is counteracted by loss of Mre11.

My only substantial comment relates to how the authors would reconcile their model for the telomeric interactions between G4 and BRCA2 with that in the study by Zimmer et al Molecular Cell 2016 (a study which is cited in passing in the manuscript, but which should be discussed in further detail given its close relationship to the current study). Zimmer et al, who also showed a combined effect of PDS treatment and BRCA2 loss on telomere stability, propose that it is the homologous recombination activities of BRCA2 that are important for telomere stability in the context of G4 formation; this would be an alternative explanation for the data in Figure 7 of the current study. Can the authors provide their thoughts on how the two different models may be reconciled?

Minor comments:

- 1) Supplementary figure 2: it would be useful to include a schematic of BRCA2 similar to that in Fig 1a, showing the locations of the B2-7, B2-8 and B2-9 fragments.
- 2) Supplementary figure 5: it is surprising that TelG-CTA is unable to bind to BRCA2OB. Does this oligonucleotide fold into G-quadruplexes under these conditions?
- 3) When discussing Fig 2b, the authors comment "These data are consistent with the notion that BRCA2 binds to telomeres during S phase when POT1 binding must be released for replication.". However, the experiment in Fig 2b does not provide information about the timing of telomere binding across the cell cycle for either protein. Instead, these data could indicate that BRCA2 has higher affinity for the variant repeats found at telomeres, particularly in ALT cells, compared to POT1.
- 4) Figure 4c-e: it's not clear if these experiments were performed in the presence or absence of BRCA2OB.

- 5) Supplementary figure 13: “heterogeneous” might be a better descriptor of the multiple FRET states of TELG5T2G4, rather than “promiscuous”, which usually refers to interactions between two entities.
- 6) Figure 5a – c: it would be useful for the reader to have diagrams of the topologies of the proposed “loop-associated G3:G4” and “tail-associated G3:G4” species that are mentioned in the text.

Reviewer #3 (Remarks to the Author):

The telomere DNA is G rich and known to have a propensity to form G-quadruplex (G4) structures. In this manuscript, the authors show that the C-terminal DNA binding domain (DBD) of tumor suppressor BRCA2 binds to G3 intermediates formed during the folding and equilibrium of G4 structures and reduce the formation of G4. Additionally, they show that the MRE11 nuclease that resects DNA double strand breaks can also resect telomeric G4 structures leading to telomere damage. Moreover, the authors show that binding of BRCA2 DBD to telomere DNA prevents MRE11-mediated G4 cleavage and telomere damage. Overall, this is an interesting study that could potentially establish another type of chromosomal locations where BRCA2 protects DNA against MRE11-mediated degradation (the first being at stalled replication forks) and maintains genome stability. However, there are a number of significant issues that need to be addressed before it can be considered for publication.

Major issues:

Figure 2e: Authors show that immunoprecipitated full-length BRCA2 can bind telomere DNA. It would be important to confirm that the C-terminal DBD is indeed responsible for the binding using a deletion mutant lacking that segment of the protein.

Figure 7: Acute Brca2 knockout may lead to perturbation of cell cycle, even with reduced p53 activity caused by SV40 large T. Were the cells sufficiently and equally synchronized? Judging by the size of nuclei shown, the cells appeared to be quite heterogenous, which may skew the results. At the minimum, cell cycle profiles should be provided. The timing may need to be fine-tuned as well.

Several BRCA2 deficient human cancer or FA cell lines exist. Author should test at least some of them and see whether their observations of telomere damage with and without MRE11 and/or PDS can be reproduced.

Other issues:

Figure 5e: Were equal amount of probe used in all lanes?

Figure 6b-e: How many times were the experiments repeated? Ideally, the full-length probe remaining should be quantified with statistics shown.

Figure 7c: Are the differences between bars 1 and 2, and bars 3 and 4, indeed not significant? How many times were the experiment repeated?

Figure S15: It would be desirable to show the inability of MRE11N-H129N to cleave TelG5GAG.

Point-by-point response to the reviewers

I am happy to resubmit this revised manuscript. Although we were first disappointed on the decision, soon we found that we were able to answer to all the concerns raised by the reviewers. During revising the manuscript according to the reviewers, we experienced that the manuscript became stronger. In this revised manuscript, we provide more insights into BRCA2 recruiting RAD51 to G4-driven stalled replication forks: BRCA2 interaction with G3-derived intermediates generated ssDNA, where RAD51 binds to. Secondly, we show that pathogenic mutations of BRCA2 OB-fold results in failure to bind telomere G4 and G3 intermediates. In this revision, we have added eight more panels in the main figure and three more in the supplementary figures. Most of the experiments were repeated several times and statistically reliable data are presented as figures. Text was revised accordingly.

We are grateful to the reviewers for helping us advance in our thinking. We learned a great deal in how BRCA2 associates with the dynamic telomere G4 and how it ensures genome integrity at the telomere by coordinating remodeling of the stalled fork and preventing it from collapse as well. Below is the detail of how we responded to the referees' questions and revised the manuscript.

To Reviewer #1

The authors studied the molecular mechanism underlying the replication fork collapse during lagging-strand telomere synthesis in which G-quadruplex (G4) structure is expected to form and BRCA2 may interact with the G4 structure. Using gel shift assay combined with single molecule FRET method, the authors demonstrate that BRCA2 associates with G-triplex (G3)-derived intermediates which is thought to be involved in the interconversion between parallel and non-

parallel G4 structures. They proposed a model where BRCA2 target such intermediate structure. They show that MRE11 nuclease G4 resecting activity is inhibited in presence of BRCA2. Using mouse strains, they also provide evidence that BRCA2 depletion results in increased telomeric damage. The study suffers from not having a clear premise, inconsistent data interpretation, poor data quality and incoherent narrative. This manuscript cannot be published in its current form.

Major points:

1. Many published studies clearly indicate that BRCA2 functions with DSS1 in associating with single stranded (not structured) DNA for the purpose of recruiting RAD51 for HR activation. Here are the supporting publications.

- *Yang et al (Science 2002) resolved a co-crystal structure of BRCA2-DBD with DSS1 bound to oligo (dT)₉ single strand DNA which clearly revealed the specific contact of OB1 and 2 domains of BRCA2-DBD with single stranded DNA bases (5-6 nucleotide) in the presence of DSS1. BRCA2-DBD-DSS1 complex, not BRCA2-DBD alone represents the cellular unit that binds single strand DNA.*

- *Zou et al (Oncogene, 2006) demonstrated that DSS1 is required for the stability of BRCA2. In this study, depletion of DSS1, similarly to the depletion of BRCA2, resulted in hypersensitivity to DNA damage, again indicating that BRCA2 does not act alone in associating with DNA.*

- *Zhao et al (Mol Cell, 2016) showed a functional significance of BRCA2-DSS1 in RPA-RAD51 mediated homologous recombination.*

- *Le et al (NAR, 2020) further verified the functional unit of BRCA2-DSS1 in HR and also showed that DSS1 and single strand DNA regulate BRCA2 oligomerization.*

Based on the previous studies, the fragment of BRCA2 the authors studied with structured (telomeric) DNA is likely biologically irrelevant.

Response: In this study, we included non-structured single stranded DNA (telomere sequence variant), where BRCA2 (BRCA2OB and BRCA2 full length) does not bind, probably without DSS1. This study started with the physiological observation that BRCA2-deficient cells exhibit telomere shortening via replication fork collapse in S phase¹. BRCA2 not only regulates HR but BRCA2 is also required to protect stalled replication forks. Jasin reports that the role of BRCA2 protecting the breakdown of stalled replication forks by BRCA2 is distinct from its dsDNA repair². In this study, we focused on the biochemical and biophysical association of BRCA2 with telomere to understand the molecular basis of telomere replication homeostasis guaranteed by the function of BRCA2. In response to the reviewer's criticism, we asked how RAD51 comes into play at the telomere G4 at later phases. In this revised manuscript, we show that the mode of BRCA2 interacting with telomere G4 provides the explanation of how BRCA2 can load RAD51 onto the ssDNA and facilitate the restart of the stalled replication forks, preventing it from resection by MRE11. In figure 7, we show that BRCA2OB binding to telomere G4 enhances the RAD51 association to ssDNA generated from the G3 intermediates. RAD51 did not compete with BRCA2 on compact telomere G4 (Fig. 7f&g). In cells, absence of Brca2 resulted in failure to load Rad51 onto the stalled forks induced by G4 stabilizer PDS or DNA Pol α inhibitor, aphidicolin (Fig. 7d). With these results, we provide insights that the recognition of telomere G4 structure by BRCA2 remodels the dynamic G4 and coordinates with RAD51-mediated restart of the stalled forks (Fig. 7h). Our study does not contradict with the previous study on BRCA2 in HR or protection of stalled replication forks. It should be appreciated that

BRCA2 is a multifunctional tumor suppressor and the role of BRCA2 does not always involve DSS1. Accumulating reports indicate that RAD51 in association with BRCA2 is more central to DNA repair and replication stress. We believe the revision with how RAD51 comes into play at telomere G4 advances our understanding on telomere homeostasis and how replication and DNA repair are linked.

2. The widely known function of BRCA2 in DNA damage and telomere damage is the homologous recombination in which BRCA2 works in conjunction with RAD51. The telomere shortening that occurs in BRCA2 deletion can also be detected in RAD51 deletion, supporting the role of both proteins in HR. Therefore, if BRCA2 association with telomeric G4 represents a functional state, it should be tested by its ability to recruit RAD51 and stimulate HR or the D-loop formation.

Response: The major problem I find with this comment is that the reviewer does not appreciate that we are working with telomere G-quadruplex, not the conventional dsDNA break. The novelty of the results come from the unique question and solid evidences provided in this manuscript. Telomere shortening can occur with many different pathways, depending on different structure of telomere. We have asked how BRCA2 associates with the telomere in S phase. It should be noted that RAD51 is present in the yeasts, where most of RAD51's functions are conserved, BRCA2 tumor suppressor is absent in yeasts. Therefore, it is reasonable to think that during the evolution of the vertebrates, the tumor suppressor BRCA2 might have acquired multiple functions³. Here, we show an interesting mechanism how telomere G4 structures are remodeled by BRCA2.

Nevertheless, we asked whether RAD51 loading onto the stalled telomere replication forks are affected by the presence or absence of BRCA2. We have already expected that RAD51 foci will be absent from these lesions when BRCA2 is absent, since numerous reports, including ours⁴.

⁵, showed that RAD51 localization to damaged sites require BRCA2. This is also true at the telomere⁶. In this revised manuscript, we showed that BRCA2 recruits RAD51 to stalled replication forks at the telomere (Fig. 7d&e). Furthermore, we showed that BRCA2 and RAD51 do not compete for the binding to telomere G4: BRCA2 binds to the G3-derived intermediates formed during compact P- and NP-G4 interconversion, while RAD51 is loaded onto the ssDNA generated by G3 at the telomere (Fig. 7f & g).

3. Related to above, since BRCA2's known role is to recruit RAD51 for HR, the conclusion from the current study, especially the simple binding (without stretching or unfolding) of BRCA2 on telomere does not make sense in terms of the known activity of BRCA2 in HR pathway.

Response: As stated above, the mode of BRCA2's interaction with telomere G4 is not the simple binding. BRCA2 preferentially recognized G3-derived intermediates, which can form during NP (non-parallel)- and P (Parallel)-G4 interconversion, without going through the unfolded single strand DNA form (Fig. 4f & 7h). When BRCA2 binds to G3 or G3:G4, it reduces the compact G4 (Fig. 2g & h, 3b & i). BRCA2 does not stretch or unfold telomere, but binding to G3 intermediates reduces G4 and generates single-stranded DNA for RAD51 to associate with (Fig. 7f & g). Taken together, the results do not contradict with the pathway known for BRCA2 at the stalled replication forks.

4. The binding affinity to telomere substrate (TelG5) is negligible, making it impossible to measure the binding constant. The higher affinity of BRCA2-DBD binding to non-telomeric G4 reveals that telomeric G4 is not a relevant substrate for BRCA2. The interaction is likely nonspecific.

Response: If the reviewer is referring to the fact that BRCA2 preferentially binds to TelG5T2G4(TTG GGG)_n or TelG5TTT (TTTGGG)_n than to TelG5 (TTAGGG)_n, I have to emphasize that they are all telomere sequences with different topologies. They are not merely representing primary sequences but we utilized them to represent different topologies of telomere G4. We have fully discussed on how they differ and why we used different telomere variants in the manuscript, in the text and in figures (Fig. 2a, Fig. 3). I would also like to emphasize that BRCA2OB is different from BRCA2-DBD. In supplementary Fig. S2, we show that the region that directly associates with telomere G4 encompasses the first OB fold to the third OB-fold, excluding the helical domain in DBD (Fig. 1a). Different telomere variants were first assessed for the binding affinity to BRCA2OB (Fig. 2, 3), then these telomere constructs were used to reflect each different dynamic states of the telomere repeats seen in G4 (ssDNA, non-parallel G4, parallel G4, G3-associated intermediates, etc.). The reviewer seems to have missed the whole idea.

5. The CD result and P/NP designation is counterintuitive and inconsistent with previous results from other laboratories.

We emphasize that the reviewer misinterpreted our CD data. To the best of our knowledge, the data is consistent with all previous publications, and it is well established in the field that the parallel folding results in a positive CD peak at ~265 nm while the antiparallel (or nonparallel) folding characteristic is represented by the peak at ~290 nm as described in the manuscript (Fig. 2d).

6. Single molecule data is of poor quality and the histogram, density plot etc. are over-fitted by

selecting center positions. As a result, the assignment of BD state is not valid in many cases.

First of all, we never attempted to fit the transition density plot (TDP): the six types of transitions (i.e., UF \rightarrow NP, NP \rightarrow UF, UF \rightarrow P, P \rightarrow UF, NP \rightarrow P, and P \rightarrow NP) are clearly visible from the TDP. Secondly, we strongly disagree with the reviewer's comment questioning the smFRET data quality. In general, the quality of smFRET data can be assessed in two aspects: FRET histograms and time trajectories. First, in the case of the FRET histogram, all data in the absence of BRCA2OB exhibit distinct population of designated FRET states (i.e., UF, NP, and P) which are well described by triple Gaussian fits. For the time trajectory, the transition density plot (Fig. 3d) clearly demonstrates that only the three FRET states (i.e., UF, NP, and P) are observed during the real-time dynamic FRET transitions, without noisy sub-population resulting from misidentified low-quality trajectories. Hence, we believe that our data confidently represent the heterogeneous and dynamic nature of telomeric G4 structures.

As the reviewer pointed out, we are aware that the FRET histograms in the presence of BRCA2OB, however, was unreliable to identify the BD state by the artificial four-state fit. In this revised manuscript, we have minimized the usage of the four-state fitting result in our downstream analysis. The purpose of displaying the over-fitted histograms in the presence of BRCA2OB (Fig. 3e & 4b in the original version) was to merely visualize that BRCA2OB induces a change in the FRET histogram particularly in the region of $E \sim 0.6$. In this revision, we replaced the over-fitted histogram to a 'differential density' histogram to explicitly show that BRCA2OB binding causes FRET shift (Fig. 2h & 3b). Differential density histogram is the direct subtraction of the measured normalized FRET histogram in the absence of BRCA2OB from that of the presence of BRCA2OB. This newly analyzed histogram unambiguously reveals that BRCA2OB mainly raises a new FRET population around $E \sim 0.6$ while depleting the three existing G4 conformational populations (UF, $E \sim 0.23$; NP, $E \sim 0.47$; and P, $E \sim 0.68$). The figures

and texts were revised accordingly.

7. There is no experimental evidence that BD occurs in transition between the two G4 folded states. The interconversion model depicted in Figure 5f is incorrect.

Response: This comment, I believe can be answered from the responses above. Also see Figure 2 and 3 in the revised figures.

8. The MRE11 activity test is of data quality that is not acceptable for publication.

Response: The activity of the purified MRE11N used in this study in comparison to MRE11N_H129N nuclease dead mutant is shown in Supplementary Fig. S13 with DAR134 and TelG5GAG, the UF ssDNA.

Figure by figure

1. In Figure 1, the EMSA gel images display that extremely small fraction of binding while majority of the labeled DNA remains unbound (thick bottom staining) in all conditions tested, suggesting that this is not specific binding. G-quadruplex, due to the folded nature of the nucleic acid has high propensity to nonspecifically interact with various proteins (Begeman, EMBO reports, 2020). By contrast, DNA binding proteins such as RPA and POT1 exhibit nanomolar affinity and POT1 displays high specific binding to telomeric single strand DNA.

Response: As we have showed and discussed throughout the study, BRCA2OB does not bind to ssDNA (TelG5GAG), single-stranded telomere DNA, but to the secondary structures like G3-

associated intermediate, which is formed while folding into telomere G4. Therefore, the amount of binding in Figure 1 can be explained. 1 to telomere has been shown, compared to BRCA2OB to telomere G4 in Fig. 2a. The significance of BRCA2 to telomere G4 association does not rely on the amount of binding *in vitro*. The impact is more on the physiological consequence of the binding, even if the amount is relatively small (Fig. 7). Binding of POT1 and BRCA2OB in EMSA to telomere variants were compared in Fig. 2b. It shows that POT1 binds to ssDNA telomere, while BRCA2OB binds to more to the telomere G4. In the binding assay, the radioactive probes shown suggest that the affinity of BRCA2OB binding to telomere G4 is not as weak as the reviewer criticizes. Binding affinity is measured in Fig. 2a.

2. How do the authors determine the dissociation constant (Kd) of BRCA2 in different G4 constructs when the Kd values must be higher than the binding fraction used in the graph? If measured Kd of BRCA2 to TelG5 is 4.4uM then why use 1uM of protein concentration during their measurements.

Response: We take the reviewer's point. Text was rewritten to tone down the claim.

3. If the authors seek to study BRCA2-DBD binding affinity comprehensively, they should test unstructured DNA such as T8-T36 used in the Yang et al's (Science 2002) structural study so that the BRCA2-DBD can be directly compared with BRCA2-DBD-DSS1.

Response: As repeatedly stated, BRCA2OB is not exactly same as BRCA2-DBD. BRCA2OB lacks the helical domain. The reviewer seems to be out-focused. I do not think that the comparison of binding constant between BRCA2OB-telomere G4 and BRCA2DBD-DSS1-ssDNA should be compared, as our study was not on the comparison of BRCA2 in ssDNA with

DSS1. Nevertheless, when we employed oligo-dA, -dC, dG, dT with BRCA2OB, we failed to observe meaningful binding, while BRCA2OB associated with telomere G4-favorable buffer. I must emphasize again that this study does not contradict with the structure shown by Yang et al (Science 2002). It is very possible that the structure of BRCA2-telomere G4 is different from BRCA2-DSS1-ssDNA.

4. In Figure 2, the binding constant is tested by EMSA. Again, due to the low binding affinity, the K_d cannot be determined accurately. Most strikingly, the telomeric substrate, TelG5 exhibits the lowest binding affinity while the non-telomeric substrates (TelG5 variants) show an improved binding. It is clear from this result that the low level of BRCA2 binding to TelG5 does not represent functional state of BRCA2 at telomere. Although the authors refer to sequences such as TTTGGG as telomere variant repeats, it is highly unlikely to encounter (GGTTTG)₄ or higher repeat number in human telomeres.

Response: Please refer to the comments above. We designed the experiments to reflect different state of topologies of telomere G4. Please understand the repeats of (TTAGGG)_n can make many conformations. We have utilized different sequence variants to reflect such differences.

5. Figure 3b top, the CD changes in low vs. high KCl is opposite of what others have shown. The parallel (260nm peak) should be stabilized by high potassium concentration. In addition, the authors mentioned ~48% molecules showed dynamics between NP-P transition of TelG5, but the dynamics may arise from the flanking sequences outside of the folded G4.

Response: Again, we emphasize that the reviewer misinterpreted our CD data. The parallel peak rises as the concentration of KCl increases from 10 mM to 100 mM (Fig. 2d).

The molecular random diffusion, in general, is much faster (<ms) than our temporal resolution (100 ms); thus, it is highly unlikely that the observed FRET dynamics originates from the random diffusion of the flanking single-stranded region. As described in our response above, the TDP analysis corroborates that only the three FRET states (UF, NP, and P) that have been consistently identified in previous reports are formed during the FRET transitions, regardless of the existence of flanking sequences. Please refer to the below for details.

6. Based on the 260 vs. 290nm distribution in CD measurement, the authors assigned FRET histogram peaks as parallel and non-parallel which is not entirely consistent with the dye positions on the DNA. The single stranded regions on both ends of the G-quadruplex likely make this interpretation difficult, also results in highly noisy fluorescence signals in individual traces.

Response: As described in the manuscript, there have been several previous studies reporting higher FRET value for the P conformation than the NP conformation^{7, 8}, as well as studies designating the lower FRET state for the P-G4. The discrepancy is due to the existence of the flanking sequence on G4 substrates. For the G4 construct without any flanking sequences, the FRET efficiency of the P conformation is lower than the NP conformational one as the reviewer mentioned, whereas G4 constructs with flanking sequences show higher FRET value for the P conformation compared to the NP conformation. We speculate that the FRET efficiency for the flanked G4 construct is likely to be more sensitive to the compactness of the G4 structure (the P-G4 has shorter inter-G-tetrad distance, i.e., more compact, than the NP-G4) as the flanking tail may exhibit diffusive positional distribution all around the G4 structure due to the random diffusion, while FRET values from G4 without flanking sequences reflect the apparent dye positions on DNA (close for NP-G4, far-diagonal for P-G4).

Since we believe that the G4 with flanking sequences on both sides is more biologically relevant

than the artificial construct having G4 isolated at the end of dsDNA, we designed our G4 constructs with flanking sequences as described in the manuscript.

The noise in the time trajectory is unlikely due to the fluctuation of the flanking region, whose random diffusion should be much faster than our temporal resolution as described right above. Moreover, the noise level of our data did not interfere with the data analysis as described in our response above. Based on these, we revised the manuscript with some more descriptions and references. Schematic of G4 constructs were revised as well to support and rationalize our FRET state assignment in the revised manuscript (Supplemental Fig. S8-S10).

7. The fitting of BD state located within the NP and P states is barely possible. It's not clear what fraction of individual traces exhibit distinguishable transitions.

Response: Prompted by the reviewer's comment and to explicitly show that the BRCA2OB binding causes FRET shift, we introduced the differential density histogram rather than fitting the BD state, as described in our response above (Fig. 2h, 3b).

For single-molecule time trajectories, we cannot clearly distinguish the NP-P-BD transition due to their close FRET values, although we see some distinct transitions right after BRCA2OB injection as illustrated in Fig. 2i. Therefore, rather than directly analyzing the BRCA2OB association and dissociation kinetics, we sought to identify the dynamic intrinsic feature of G4 constructs that correlates with their BRCA2OB binding capability, which led to the discovery of BRCA2OB binding towards G3 intermediates (also refer to the below figure-by-figure point #11).

8. Author used the TDP to show the transition between the states of UF, NP, and P. TDP for TelG5TAA and TelG5 is not reported. There is a discrepancy as the histogram doesn't show any

unfolded population for TelG5 but the TDP showed the major transition from unfolded state. Author should clarify this discrimination.

Response: The reviewer missed the points in our manuscript. TDPs for TelG5TAA and TelG5 are unambiguously displayed in the upper panel of Fig. 3d. Moreover, transition density has nothing to do with relative population of each FRET state because only the transitioning moments are collected to plot the TDP. Our result shows that although the UF conformation does not form a significant population, the structural transition between the two major, folded states (NP and P) predominantly occurs via transient complete unfolding (passing through the UF state) for TelG5 and TelG5TAA, as described in our manuscript. This is not a discrepancy. It is the beauty of single-molecule analysis that enables identification of short-lived intermediate states regardless of its population at equilibrium.

9. Author mentioned that “Notably, the density of the NP-P direct transition matched the relative binding affinity of BRCA2OB to G4 substrates (Fig. 4f).” However, the relative BRCA2 binding is only ~10-20% from EMSA whereas ~60% same binding from smFRET measurement for TelG5TTT and TelG5TT.

Response: Again, the reviewer misinterpreted our data. The relative BRCA2 binding refers to the relative binding strength of BRCA2OB towards TelG5 (not the percent). We added description in the figure legend to clarify this point.

10. Figure 4, the BD state assignment, again is not clear, especially for TelG5TAA and TelG5. Figure 4e shows TelG5TTT trace, yet no BD state is shown.

Response: As described in our response to the point #6, we replaced the figure to the differential density histogram to explicitly show that the BRCA2OB binding induces FRET populational

shift towards $E \sim 0.6$.

Fig. 3e is a representative time trajectory of TelG5TTT alone in the absence of BRCA2OB, exemplifying the direct transition between the NP and P conformations, which is indicated by the quantitative TDP analysis (Fig. 3d). We added detailed description in the figure legends to clarify whether each data was obtained in the absence or the presence of BRCA2OB.

11. There is no clear evidence that BRCA2 binds to intermediate between NP and P. If the traces show clear transition between P, NP and BD, then the authors can analyze the partition between P to BD vs. NP to BD.

Response: Unfortunately, we were not able to clearly discriminate the P, NP, and BD states in time trajectories due to their close FRET values. Nonetheless, this is the first discovery of a new molecular interaction mode of BRCA2 towards the G4 substrates, which shows the biological impact. Comprehensive investigations of the biophysical structure and mechanism of BRCA2-G4 (and -G3) interaction are warranted.

12. Authors propose a model for NP-P transition during which BRCA2 engages. The propose model of NP-G4 to P-G4 transition at Fig. 5f where the fourth strand remains in the same orientation i.e unchanged. According to the FRET peak, the fourth strand must be oriented differently to induce a different FRET state. This model is misleading.

Response: As described in the figure-by-figure point #6, the FRET efficiency for G4 constructs with flanking sequences is not predictable merely from the dye orientation; thus, the relative compactness of the G4 structure rather than the orientation of the fourth strand is likely to determine the FRET efficiency for our G4 substrates. We believe that the NP-G4 conformation to be the hybrid structure, because it is the major population that is captured by bulk structural

experiments. To better reflect the FRET efficiency of each G4 conformation, we slightly revised the cartoon in the figure.

13. In Figure 5, the G-triplex diagram does not match the FRET positions. The high fraction of low FRET indicates mostly unfolded state of the DNA. Again, the binding is not significant as shown on the gel.

Response: We added a molecular cartoon that matches each FRET state (UF, G3 & loop-associated G3:G4, and tail-associated G3:G4) in Fig. 4b. The diagram in Fig. 4a summarizes the result that BRCA2OB binds to the G3-derived intermediates (not to the UF conformation).

For the binding affinity, as described in our response to the figure-by-figure point #2, we mainly focused on the binding strength relative to TelG5 because the absolute value of K_d may be different between the BRCA2OB fragment and the full-length BRCA2.

To Reviewer #2

In this study, the authors provide compelling biochemical evidence for the ability of BRCA2 to interact with G4-forming telomeric DNA and perturb its structure, most likely by binding to folding intermediates between different conformations of G4. They also demonstrate that Mre11 is able to cleave DNA containing G4, and this ability is blocked by BRCA2. The experimental data are robust, and include complementary lines of evidence from both EMSA and single molecule FRET analyses. The FRET analyses, in particular, allow the authors to probe the molecular interactions between BRCA2 and G4 in some detail. Finally, a small amount of in vivo analysis in BRCA2-deficient mouse cells supports a synergistic effect of BRCA2 loss and G4

stabilization on telomeric DNA damage, which is counteracted by loss of Mre11.

Response: We thank the reviewer for appreciating our work.

My only substantial comment relates to how the authors would reconcile their model for the telomeric interactions between G4 and BRCA2 with that in the study by Zimmer et al Molecular Cell 2016 (a study which is cited in passing in the manuscript, but which should be discussed in further detail given its close relationship to the current study). Zimmer et al, who also showed a combined effect of PDS treatment and BRCA2 loss on telomere stability, propose that it is the homologous recombination activities of BRCA2 that are important for telomere stability in the context of G4 formation; this would be an alternative explanation for the data in Figure 7 of the current study. Can the authors provide their thoughts on how the two different models may be reconciled?

Response: We thank the reviewer for highly appreciating our manuscript. Regarding the study from Zimmer et al., I don't think this study contradicts with the paper. In fact, this manuscript is in agreement with Zimmer et al. In Figure 1f, we showed that preincubation of the TelG5 with PDS results in the decrease of BRCA2 binding to telomere G4. This result supports that BRCA2 binds less to stabilized G4, resulting from PDS treatment, and that BRCA2 binds to G3-associated intermediates much better. Therefore, PDS treatment will increase DNA breaks (as seen in Fig. 7a,& b). Notably, PDS treatment at telomeres is effective in Brca2-deficient cells, compared to Brca2-intact cells (Fig. 7a, b). This result is consistent with the observation from Zimmer et al.

BRCA2OB does not possess RAD51-binding region of BRCA2. When we included BRCA2OB

and RAD51 in EMSA, we observed that the BRCA2OB and RAD51 do not compete for each other at the telomere. Interestingly, telomere G4 was recognized and preferentially bound by BRCA2 and RAD51 association to telomere increased in the presence of BRCA2OB (Fig. 7f & g). As RAD51 localization to stalled fork-containing telomere required BRCA2 (Fig. 7a, e), it is conceivable to think that BRCA2OB association to G3-derived intermediates generates ssDNA telomere that RAD51 can associate to. This way, BRCA2 achieves two tasks: recruitment of RAD51 to stalled forks and loads it for fork restart; BRCA2-bound G3 intermediates reduces G4 that is attacked by MRE11 (Fig. 7a, b). Therefore, BRCA2 works as a scaffold (platform) to remodel telomere G4, and also coordinate HR by recruiting RAD51.

Minor comments:

1) *Supplementary figure 2: it would be useful to include a schematic of BRCA2 similar to that in Fig 1a, showing the locations of the B2-7, B2-8 and B2-9 fragments.*

Response: It is done as suggested in the revised manuscript (supplementary Fig. 2a).

2) *Supplementary figure 5: it is surprising that TelG-CTA is unable to bind to BRCA2OB. Does this oligonucleotide fold into G-quadruplexes under these conditions?*

Response: We have included the smFRET data of TelG5-CTA in Supplementary Fig. S12. It shows that it does form G4. However, it forms a very stable conformation, varying from the telomere variant sequences (Supplementary Fig. S12). This result confirms that BRCA2OB binds to G3-associated intermediate conformation rather than stable telomere G4.

3) When discussing Fig 2b, the authors comment “These data are consistent with the notion that BRCA2 binds to telomeres during S phase when POT1 binding must be released for replication.”. However, the experiment in Fig 2b does not provide information about the timing of telomere binding across the cell cycle for either protein. Instead, these data could indicate that BRCA2 has higher affinity for the variant repeats found at telomeres, particularly in ALT cells, compared to POT1.

Response: The reviewer’s point is well taken. The text is revised as follows: The results indicate that BRCA2 and POT1 bind to distinct structures of telomere. The reviewer’s points are also discussed in the revised manuscript: Whether this mode of BRCA2-telomere binding is associated with the prevention or induction of ALT should be studied in the future.

4) Figure 4c-e: it’s not clear if these experiments were performed in the presence or absence of BRCA2OB

Response: It was done in the absence of BRCA2. This was made clear in the figure legend in the revised version.

5) Supplementary figure 13: “heterogeneous” might be a better descriptor of the multiple FRET states of TELG5T2G4, rather than “promiscuous”, which usually refers to interactions between two entities.

Response: Thank you for the comment. Text is revised accordingly.

6) Figure 5a – c: it would be useful for the reader to have diagrams of the topologies of the

proposed “loop-associated G3:G4” and “tail-associated G3:G4” species that are mentioned in the text.

Response: In response, diagrams are included (Fig.4b). Thanks for the suggestion.

To Reviewer #3

The telomere DNA is G rich and known to have a propensity to form G-quadruplex (G4) structures. In this manuscript, the authors show that the C-terminal DNA binding domain (DBD) of tumor suppressor BRCA2 binds to G3 intermediates formed during the folding and equilibrium of G4 structures and reduce the formation of G4. Additionally, they show that the MRE11 nuclease that resects DNA double strand breaks can also resect telomeric G4 structures leading to telomere damage. Moreover, the authors show that binding of BRCA2 DBD to telomere DNA prevents MRE11-mediated G4 cleavage and telomere damage. Overall, this is an interesting study that could potentially establish another type of chromosomal locations where BRCA2 protects DNA against MRE11-mediated degradation (the first being at stalled replication forks) and maintains genome stability. However, there are a number of significant issues that need to be addressed before it can be considered for publication.

Major issues:

Figure 2e: Authors show that immunoprecipitated full-length BRCA2 can bind telomere DNA. It would be important to confirm that the C-terminal DBD is indeed responsible for the binding using a deletion mutant lacking that segment of the protein.

Response: The experiment was performed in HeLa cell, which expressed BAC harboring full length BRCA2 with GFP-tag (Fig. 6b). However, the reviewer’s point is well taken. In

response, 2X MBP tagged construct expressing BRCA2 OB-fold mutations (D2723H; R2973C), found from human cancer patients, were generated by site-directed mutagenesis from the wild-type, transfected into 293T cells along with the wild-type control. Instead of the C-terminus truncation, we used point mutants found from cancer patients, as these would be more physiological. The result in Fig. 6d showed that the OB mutant D2723H; R2973C was incapable of binding to telomere G4, particularly to G3 intermediate (Fig. 6d). In addition, recombinant BRCA2OB-D2723H and -R2973C were tested for their ability to bind to telomere variants in EMSA. The result showed that OB-fold mutants failed to bind to telomere G4 and G3-derived structures, while wild-type BRCA2OB consistently exhibited preferential binding towards TelG5-G3 and TelG5-TTT (Fig. 6a). These results indicate that BRCA2OB-fold mutations that are not able to bind to telomere G4 is pathogenic. Note that the telomere G4-binding region of BRCA2 lacks helical domain in BRCA2-DBD (Fig. 1). I hope these experiments and the results satisfy the reviewer.

Figure 7: Acute Brca2 knockout may lead to perturbation of cell cycle, even with reduced p53 activity caused by SV40 large T. Were the cells sufficiently and equally synchronized? Judging by the size of nuclei shown, the cells appeared to be quite heterogenous, which may skew the results. At the minimum, cell cycle profiles should be provided. The timing may need to be fine-tuned as well.

Response: MEFs are hard to synchronize due to its heterogeneity. We tried our best to synchronize the cells. However, the reviewer is right that depending on the efficiency of the synchronization, the effect might be different. In this revised manuscript, we included aphidicholin treatment, which markedly increases the stalling of replication forks in S phase as it is an inhibitor of DNA Pol α . Also, we carefully repeated the experiment and replaced the

figures. More than 120 cells each were scored and statistical measurements (mean \pm s.e.m.) with p values are shown to avoid skewing of the result.

Several BRCA2 deficient human cancer or FA cell lines exist. Author should test at least some of them and see whether their observations of telomere damage with and without MRE11 and/or PDS can be reproduced.

Response: The reviewer's point is well taken. Unfortunately, we could not get hold of FA cell lines that is deficient in BRCA2. However, we believe that the isogenic experiment with the conditional mouse MEFs (Fig. 7a-d) provide solid molecular evidence. In Figure 7, we have assessed the effect of Mre11, PDS, and aphidicholin in telomere damage in the presence or absence of Brca2 as well. Moreover, we assessed the Rad51 localization to stalled forks at the telomere as well. The reason we stick to the conditional Brca2 MEFs is because we have accumulated experiences to abrogate Brca2 in an efficient way in a relative short time and perform the assay. As we scored good numbers of cells (n>120 each) with statistics, we believe the data are reliable. In this revision, we provide evidence that the pathogenic point mutations at the OB-fold of BRCA2 (Fig. 6) abrogates its binding to telomere G4, particularly to G3-derived intermediates (Fig. 6d). With these additional results, I hope the reviewer is satisfied.

Other issues:

Figure 5e: Were equal amount of probe used in all lanes?

Response: Yes. The result is the representation from two independent experiments.

Figure 6b-e: How many times were the experiments repeated? Ideally, the full-length probe

remaining should be quantified with statistics shown.

Response: The experiment was repeated more than three times. Statistics is with p values are included in the revised manuscript.

Figure 7c: Are the differences between bars 1 and 2, and bars 3 and 4, indeed not significant? How many times were the experiment repeated?

Response: This comment relates to the fact that MEFs are hard to synchronize. However, we have repeated the experiment three times. Experiments including aphidicholin were included and the experiments were repeated. In the revised figure, more than 120 cells each were scored and the exact *p* values are shown.

Figure S15: It would be desirable to show the inability of MRE11N-H129N to cleave TelG5GAG.

Response: Thank you for the comment. The result is included as Supplementary Fig. S13 in the revised manuscript. MRE11N-H129N does not cleave TelG5GAG.

References

1. Min, J. *et al.* The breast cancer susceptibility gene BRCA2 is required for the maintenance of telomere homeostasis. *J Biol Chem* **287**, 5091-5101 (2012).
2. Schlacher, K. *et al.* Double-strand break repair-independent role for BRCA2 in blocking stalled replication fork degradation by MRE11. *Cell* **145**, 529-542 (2011).
3. Lee, H. Cycling with BRCA2 from DNA repair to mitosis. *Exp Cell Res* **329**, 78-84 (2014).

4. Min, J., Park, P.G., Ko, E., Choi, E. & Lee, H. Identification of Rad51 regulation by BRCA2 using *Caenorhabditis elegans* BRCA2 and bimolecular fluorescence complementation analysis. *Biochem Biophys Res Commun* **362**, 958-964 (2007).
5. Kwon, M.S. *et al.* Brca2 abrogation engages with the alternative lengthening of telomeres via break-induced replication. *FEBS J* **286**, 1841-1858 (2019).
6. Badie, S. *et al.* BRCA2 acts as a RAD51 loader to facilitate telomere replication and capping. *Nat Struct Mol Biol* **17**, 1461-1469 (2010).
7. Tippana, R., Hwang, H., Opresko, P.L., Bohr, V.A. & Myong, S. Single-molecule imaging reveals a common mechanism shared by G-quadruplex-resolving helicases. *Proc Natl Acad Sci U S A* **113**, 8448-8453 (2016).
8. Ray, S., Bandaria, J.N., Qureshi, M.H., Yildiz, A. & Balci, H. G-quadruplex formation in telomeres enhances POT1/TPP1 protection against RPA binding. *Proc Natl Acad Sci U S A* **111**, 2990-2995 (2014).

REVIEWERS' COMMENTS

Reviewer #1 (Remarks to the Author):

The authors have reasonably addressed the issues raised in the original reviews. The major concern still remains regarding the physiological relevance of this finding. Compared to the sequence specific and exquisite binding of POT1 with the K_d on the order of 1-10nM, BRCA2 binding demonstrated by the authors is extremely weak with the K_d of 1-4 μ M. Based on the known function of BRCA2, it does not make sense that BRCA2 will interact with the naked telomeric overhang with a high selectivity. It's not clear if the BRCA2 accumulation at the telomere is happening in the context of fully bound shelterin proteins, partially bound or not bound at all. The accumulation of BRCA2 may arise from a secondary effect of other factors such as tankyrase or shelterin proteins. Despite some caveats, as authors emphasize, the BRCA2 binding preference to G4 is a new information. Future structural studies will be needed to test this hypothesis.

Reviewer #2 (Remarks to the Author):

The authors have included additional data that I believe strengthens the manuscript; in particular, the addition of data regarding RAD51 binding to telomeric DNA, both in vitro and in cells, provides a useful extension to their model.

However, I am disappointed that they did not add any discussion of the previous paper by Zimmer et al 2016 to the manuscript, as I requested. That study was the first to demonstrate the importance of BRCA2 to replication of G4-containing DNA, including telomeres, so their contribution should be acknowledged in the current manuscript.

Reviewer #3 (Remarks to the Author):

In this resubmitted manuscript, the authors have experimentally addressed several key concerns raised in the first round of review and clarified a number of other points. Overall, the manuscript is significantly improved in my view, although there are still some issues that need to be addressed.

With respect to my own comments, the authors largely addressed the first major concern by adding a new experiment showing that two clinically relevant BRCA2 variants in the OB folds substantially reduced telomeric G4 binding (Figure 6). This is certainly a strong result; however, there is no explanation, or even speculation, of how the two variants could affect the binding. This needs to be discussed, especially as the structure of this part of BRCA2 is available. Interestingly, one of the variants, D2723H, has been shown to unmask a nuclear export signal in BRCA2 and leads to cytoplasmic localization. Are the two effects connected? Also, a gel image of the purified wt and variant BRCA2OB fragments should be shown.

My second and third major concerns are not addressed very well. Addition of aphidicolin is helpful, but I would still prefer to see some key experiments done in human cells and with better synchronization. The authors say they were unable to obtain human BRCA2 mutant FA cells, but there are also mutant cancer cells available. At the very least, an siRNA knockdown should be doable, and MRE11 inhibition can be done with mirin.

In response to reviewer #1, the authors clarified that telomeric G4 binding by BRCA2OB and general DNA binding by BRCA2DBD are likely separable activities, one requiring DSS1 and one not. This is a reasonable proposition to me. To fully address this issue, it would be best if the authors could purify the whole DBD, including both the helical domain and the OB folds, test telomeric G4 vs generic DNA binding with and without DSS1, and compare the binding activities with OB alone.

Minor points:

Figure 1a: BRCA2 should be 3,418aa not 3,419aa.

Page 5: The molecular weight of BRCA2 is about 384KD not ~420KD.

Point-by-point response to the reviewers

To Reviewer #1:

The authors have reasonably addressed the issues raised in the original reviews. The major concern still remains regarding the physiological relevance of this finding. Compared to the sequence specific and exquisite binding of POT1 with the K_d on the order of 1-10nM, BRCA2 binding demonstrated by the authors is extremely weak with the K_d of 1-4 μ M. Based on the known function of BRCA2, it does not make sense that BRCA2 will interact with the naked telomeric overhang with a high selectivity.

Response: The reviewer seems slightly misunderstood. We have shown that BRCA2 interacts with telomere G4, which is the stacks of planar G-tetrads. Particularly, BRCA2 associates with G3 intermediates that form during interconversion between parallel and non-parallel G4. The portion of the G3 intermediates is much less, compared to ssDNA telomere. With compelling lines of evidence, we showed that BRCA2 does not bind to naked ssDNA telomere. Therefore, we do not claim that BRCA2 binds to naked telomeric overhang. However, G4 can form during telomere replication. In this line, direct comparison of the K_d of POT1 vs BRCA2 might not be fair: POT1 binds to G-rich ssDNA telomere repeats with high affinity, whereas BRCA2 specifically binds to G3 intermediates in dynamic telomere G4 but not to the ssDNA. Ray and colleagues reported that G4-forming property of telomere overhang enhances POT1/TPP1 protection and prevents from DNA damage-inducing RPA binding to ssDNA³. Furthermore, accumulating data suggest that POT1/TPP1 destabilizes telomere G4 in high concentration, whereas in low concentration the complex captures G4⁴. Therefore, both POT1 and BRCA2 OB-folds recognize telomere G4 and bind to it. However, POT1 destabilizes G4, whereas BRCA2 binds to the G3 intermediates generated during interconversion of NP- and P-G4. I would like to emphasize that the mode of BRCA2 binding to telomere G4

is different from that of POT1. We believe that it is reflected in the K_d value. In response to the reviewer, differential binding property of POT1 and BRCA2 is further discussed and the reference for how POT1 specifically binds to telomere ssDNA is included in the revised manuscript in p23 as follows: “Notably, the G4-forming property of telomere overhang enhances POT1/TPP1 protection and prevents from DNA damage-inducing RPA to bind to telomeres³. Furthermore, POT1/TPP1 first captures telomere G4 but increasing amount of POT1 destabilizes G4⁴, facilitating T-loop formation. The differential binding mode of BRCA2 and POT1 to telomeres (Fig. 2b) suggest that BRCA2 functions at telomere G4 exclusively during replication, but less likely to be found at the T-loop. In this line of thinking, the reason why G-richness is conserved throughout eukaryotic evolution can be explained. First, G4-forming property prevents telomere recombination through capturing POT1/TPP1. Secondly, by adopting dynamic nature, compared to G4 found at the promoter region, e.g., *c-MYC* promoter, and the ability of BRCA2 to associate with the dynamic telomere G4, the mission of maintaining telomere sequence, and G4, during replication can be achieved. Our results unravel the long-standing question of how specific telomere sequences are linked to telomere function and contribute to genome integrity. And, the tumor suppressor BRCA2 is essential for telomere integrity.”

It's not clear if the BRCA2 accumulation at the telomere is happening in the context of fully bound shelterin proteins, partially bound or not bound at all. The accumulation of BRCA2 may arise from a secondary effect of other factors such as tankyrase or shelterin proteins. Despite some caveats, as authors emphasize, the BRCA2 binding preference to G4 is a new information. Future structural studies will be needed to test this hypothesis.

Response: The data from the differential binding of POT1 vs BRCA2 suggests that BRCA2-bound telomere and shelterin bound telomere differ in their topology. The likelihood is that

BRCA2 does not bind to telomeres in the context of fully bound shelterin proteins as discussed above⁵: 1) POT1 binds and then disrupts G4, whereas BRCA2 binds to G3 intermediates while interconverting between parallel and non-parallel G4; 2) BRCA2 binding to telomere is restricted to S phase to control G4 for telomere replication homeostasis. Both T-loops and G4 must be dismantled to permit efficient telomere replication, and Boulton has shown that RTEL unwinds T-loops while replication⁶. Here, we showed that BRCA2 binding to telomeric G4 remodels it and allows the restart of G4-driven stalled replication forks. We added this information into the discussion.

To Reviewer #2:

The authors have included additional data that I believe strengthens the manuscript; in particular, the addition of data regarding RAD51 binding to telomeric DNA, both in vitro and in cells, provides a useful extension to their model.

Response: We thank the reviewer for appreciating our work.

However, I am disappointed that they did not add any discussion of the previous paper by Zimmer et al 2016 to the manuscript, as I requested. That study was the first to demonstrate the importance of BRCA2 to replication of G4-containing DNA, including telomeres, so their contribution should be acknowledged in the current manuscript.

Response: We apologize for not fully acknowledging Zimmer and colleagues' work (2016). The paper was cited but with mere mistake, it wasn't discussed fully. In the revised manuscript, we discussed on the lethality of PDS treatment in BRCA2-mutated cells as follows in p24: "This observation is in agreement with the previous report by Zimmer and colleagues, which showed

that PDS treatment increased telomere damage and reduced the viability of cells lacking BRCA2. These results altogether imply that G4-stabilizing drugs have strong therapeutic potential in BRCA2-deficient tumors⁵.”

To Reviewer #3:

In this resubmitted manuscript, the authors have experimentally addressed several key concerns raised in the first round of review and clarified a number of other points. Overall, the manuscript is significantly improved in my view, although there are still some issues that need to be addressed.

With respect to my own comments, the authors largely addressed the first major concern by adding a new experiment showing that two clinically relevant BRCA2 variants in the OB folds substantially reduced telomeric G4 binding (Figure 6). This is certainly a strong result; however, there is no explanation, or even speculation, of how the two variants could affect the binding. This needs to be discussed, especially as the structure of this part of BRCA2 is available. Interestingly, one of the variants, D2723H, has been shown to unmask a nuclear export signal in BRCA2 and leads to cytoplasmic localization. Are the two effects connected? Also, a gel image of the purified wt and variant BRCA2OB fragments should be shown.

Response: We thank the reviewer for appreciating our work. Speculations regarding the structural alteration by the mutation of D2723H and R2973C are incorporated in the revised manuscript, p25, as follows:

“The two mutants used in this study, D2723H and R2973C, are located at OB1 and OB2, respectively. A report has shown that D2723H affects a key Asp residue in OB1, which is involved in hydrogen bonding with other residues in OB1 to stabilize helical domain (HD)/OB1

interface, hence the mutant D2723H exhibits high Gibbs free energy, implying destabilization of the protein folding. Notably, D2723H mutation results in the disruption of protein conformation⁷. The D2723H mutation also interferes with DSS1 binding, unmasking the nuclear export signal (NES), hence rendering the protein cytosolic⁸. As our experiments were on direct binding assay and nuclease assay with telomere variants (Fig. 6), the defective interaction with telomere G4 is likely to result from the structural alteration rather than from defective localization of the protein. The fact that we found B2-8, which contains OB2 and OB3 but lacks OB1, associated with telomere G4 (Supplementary Fig. 2) supports this notion. Nevertheless, whether DSS1 is recruited to the ssDNA region of G3 intermediates, when BRCA2 binds, and contributes to the restart of G4-driven stalled replication forks remains to be elucidated. In comparison, R2973C is known to be less pathogenic and the information regarding R2973C is limited. It is possible that the charged Arg substituted to neutral Cys interferes with the association with DNA. When we utilized the structure prediction software from RoseTTAFold², D2723H and R2973C both displayed conformational change, particularly at the tower domain (see Figure 1 below), suggesting that the two mutations at the OB fold interrupt with the structure that associates with telomere G4. Revealing the structure of BRCA2-telomere G4 complex will answer to how these two mutations affect BRCA2 binding to telomere G4.”

Figure 1. Predicted structure of D2723H and R2973C, overlaid with published mouse Brca2 structure by Yang et al. (2002)¹, using RoseTTAFold².

Gel image of purified WT and mutants BRCA2OB is included at right in Figure 6a in the revised manuscript.

My second and third major concerns are not addressed very well. Addition of aphidicolin is helpful, but I would still prefer to see some key experiments done in human cells and with better synchronization. The authors say they were unable to obtain human BRCA2 mutant FA cells, but there are also mutant cancer cells available. At the very least, an siRNA knockdown should be doable, and MRE11 inhibition can be done with mirin.

Response: In response, we utilized siRNA experiments in HeLa cells. As shown below, the results were consistent with the mouse TBI fibroblasts we used in Figure 7. Briefly, depletion of BRCA2 resulted in marked increase of damaged telomere, comparable to the level of PDS

treatment in cells intact with BRCA2. Knockdown of MRE11 compromised the telomeric damage, induced by the depletion of BRCA2. When PDS was added to BRCA2-depleted HeLa cells, the damage was aggravated, which was again relieved by knockdown of MRE11, consistent with Figure 7. Number of cells (n) analyzed are marked below. Please understand that we did not include this figure in the manuscript because Nature Comm does not require siRNA experiments. Synchronization in HeLa cells are easier, compared to MEFs. However, in the main figure with MEFs, we scored more than 150 cells and the p values are marked to avoid the problem of synchronization difficulty, compared to human cells.

Mirin experiment was included in the Supplementary Figure 15. The result is consistent with the depletion of Mre11 with shMre11. This was discussed in the revised text.

Figure 1. HeLa cells were synchronized at G1/S with double thymidine block for 16 h, then released into the cell cycle with several washes. siRNA for *BRCA2* or *MRE11* for transfected 6 h prior to thymidine block. Four hours post release, PDS treatment was done then fixed for immunofluorescence assay with anti-TRF1 and anti-gamma-H2AX. N, number of cells analyzed. p values were obtained by the students's t test.

In response to reviewer #1, the authors clarified that telomeric G4 binding by BRCA2OB and general DNA binding by BRCA2DBD are likely separable activities, one requiring DSS1 and one not. This is a reasonable proposition to me. To fully address this issue, it would be best if the authors could purify the whole DBD, including both the helical domain and the OB folds, test telomeric G4 vs generic DNA binding with and without DSS1, and compare the binding activities with OB alone.

Response: We thank the reviewer for appreciating our work. The reviewer's points are well taken. However, the suggested experiments are exactly our next structural studies. I would appreciate if you could understand that the amount of works needed to exactly address this issue requires a long way to go, which is to be our potential next paper. For your information, we have purified DBD, which binds to telomere G4 variants, but it does not bind anonymous ssDNA or TelG5GAG (ssDNA).

Minor points:

Figure 1a: BRCA2 should be 3,418aa not 3,419aa.

Response: We thank the reviewer for correction. It is corrected in Figure 1 and supplementary Figure 2.

Page 5: The molecular weight of BRCA2 is about 384KD not ~420KD.

Response: The molecular weight of BRCA2 is stated differently from author to author (S. Aaronson, ~400 kDa; Venkitaraman ~384 kDa; S. West, 384 kDa), including the companies that sell antibodies (some companies state ~400 kDa). In response, we changed it to ~384 kDa.

References

1. Yang, H. *et al.* BRCA2 function in DNA binding and recombination from a BRCA2-DSS1-ssDNA structure. *Science* **297**, 1837-1848 (2002).
2. Baek, M. *et al.* Accurate prediction of protein structures and interactions using a three-track neural network. *Science* **373**, 871-876 (2021).
3. Ray, S., Bandaria, J.N., Qureshi, M.H., Yildiz, A. & Balci, H. G-quadruplex formation in telomeres enhances POT1/TPP1 protection against RPA binding. *Proc Natl Acad Sci U S A* **111**, 2990-2995 (2014).
4. Xu, M. *et al.* Active and Passive Destabilization of G-Quadruplex DNA by the Telomere POT1-TPP1 Complex. *J Mol Biol* **433**, 166846 (2021).
5. Zimmer, J. *et al.* Targeting BRCA1 and BRCA2 Deficiencies with G-Quadruplex-Interacting Compounds. *Mol Cell* **61**, 449-460 (2016).
6. Vannier, J.B., Pavicic-Kaltenbrunner, V., Petalcorin, M.I., Ding, H. & Boulton, S.J. RTEL1 dismantles T loops and counteracts telomeric G4-DNA to maintain telomere integrity. *Cell* **149**, 795-806 (2012).
7. Lee, M., Shorthouse, D., Mahen, R., Hall, B.A. & Venkitaraman, A.R. Cancer-causing BRCA2 missense mutations disrupt an intracellular protein assembly mechanism to disable genome maintenance. *Nucleic Acids Res* **49**, 5588-5604 (2021).
8. Jeyasekharan, A.D. *et al.* A cancer-associated BRCA2 mutation reveals masked nuclear export signals controlling localization. *Nat Struct Mol Biol* **20**, 1191-1198 (2013).